# Optimal Rates for Random Fourier Features

**Bharath K. Sriperumbudur**[*]
Department of Statistics
Pennsylvania State University
University Park, PA 16802, USA
bks18@psu.edu

**Zoltán Szabó**[*]
Gatsby Unit, CSML, UCL
Sainsbury Wellcome Centre, 25 Howland Street
London - W1T 4JG, UK
zoltan.szabo@gatsby.ucl.ac.uk

## Abstract

Kernel methods represent one of the most powerful tools in machine learning to tackle problems expressed in terms of function values and derivatives due to their capability to represent and model complex relations. While these methods show good versatility, they are computationally intensive and have poor scalability to large data as they require operations on Gram matrices. In order to mitigate this serious computational limitation, recently randomized constructions have been proposed in the literature, which allow the application of fast linear algorithms. Random Fourier features (RFF) are among the most popular and widely applied constructions: they provide an easily computable, low-dimensional feature representation for shift-invariant kernels. Despite the popularity of RFFs, very little is understood theoretically about their approximation quality. In this paper, we provide a detailed finite-sample theoretical analysis about the approximation quality of RFFs by (i) establishing optimal (in terms of the RFF dimension, and growing set size) performance guarantees in uniform norm, and (ii) presenting guarantees in $L^r$ ($1 \leq r < \infty$) norms. We also propose an RFF approximation to derivatives of a kernel with a theoretical study on its approximation quality.

## 1 Introduction

Kernel methods [17] have enjoyed tremendous success in solving several fundamental problems of machine learning ranging from classification, regression, feature extraction, dependency estimation, causal discovery, Bayesian inference and hypothesis testing. Such a success owes to their capability to represent and model complex relations by mapping points into high (possibly infinite) dimensional feature spaces. At the heart of all these techniques is the kernel trick, which allows to *implicitly* compute inner products between these high dimensional feature maps, $\lambda$ via a kernel function $k$: $k(\mathbf{x}, \mathbf{y}) = \langle \lambda(\mathbf{x}), \lambda(\mathbf{y}) \rangle$. However, this flexibility and richness of kernels has a price: by resorting to implicit computations these methods operate on the Gram matrix of the data, which raises serious computational challenges while dealing with large-scale data. In order to resolve this bottleneck, numerous solutions have been proposed, such as low-rank matrix approximations [25, 6, 1], explicit feature maps designed for additive kernels [23, 11], hashing [19, 9], and random Fourier features (RFF) [13] constructed for shift-invariant kernels, the focus of the current paper.

RFFs implement an extremely simple, yet efficient idea: instead of relying on the implicit feature map $\lambda$ associated with the kernel, by appealing to Bochner's theorem [24]—any bounded, continuous, shift-invariant kernel is the Fourier transform of a probability measure—-[13] proposed an explicit low-dimensional random Fourier feature map $\phi$ obtained by empirically approximating the Fourier integral so that $k(\mathbf{x}, \mathbf{y}) \approx \langle \phi(\mathbf{x}), \phi(\mathbf{y}) \rangle$. The advantage of this explicit low-dimensional feature representation is that the kernel machine can be efficiently solved in the primal form through fast linear solvers, thereby enabling to handle large-scale data. Through numerical experiments, it has also been demonstrated that kernel algorithms constructed using the approximate kernel do not

---

[*]Contributed equally.

suffer from significant performance degradation [13]. Another advantage with the RFF approach is that unlike low rank matrix approximation approach [25, 6] which also speeds up kernel machines, it approximates the entire kernel function and not just the kernel matrix. This property is particularly useful while dealing with out-of-sample data and also in online learning applications. The RFF technique has found wide applicability in several areas such as fast function-to-function regression [12], differential privacy preserving [2] and causal discovery [10].

Despite the success of the RFF method, surprisingly, very little is known about its performance guarantees. To the best of our knowledge, the only paper in the machine learning literature providing certain theoretical insight into the accuracy of kernel approximation via RFF is [13, 22]:[1] it shows that $A_m := \sup\{|k(\mathbf{x}, \mathbf{y}) - \langle \phi(\mathbf{x}), \phi(\mathbf{y}) \rangle_{\mathbb{R}^{2m}}| : \mathbf{x}, \mathbf{y} \in \mathcal{S}\} = O_p(\sqrt{\log(m)/m})$ for any compact set $\mathcal{S} \subset \mathbb{R}^d$, where $m$ is the number of random Fourier features. However, since the approximation proposed by the RFF method involves empirically approximating the Fourier integral, the RFF estimator can be thought of as an *empirical characteristic function* (ECF). In the probability literature, the systematic study of ECF-s was initiated by [7] and followed up by [5, 4, 27]. While [7] shows the almost sure (a.s.) convergence of $A_m$ to zero, [5, Theorems 1 and 2] and [27, Theorems 6.2 and 6.3] show that the optimal rate is $m^{-1/2}$. In addition, [7] shows that almost sure convergence cannot be attained over the entire space (i.e., $\mathbb{R}^d$) if the characteristic function decays to zero at infinity. Due to this, [5, 27] study the convergence behavior of $A_m$ when the diameter of $\mathcal{S}$ grows with $m$ and show that almost sure convergence of $A_m$ is guaranteed as long as the diameter of $\mathcal{S}$ is $e^{o(m)}$. Unfortunately, all these results (to the best of our knowledge) are asymptotic in nature and the only known finite-sample guarantee by [13, 22] is non-optimal. In this paper (see Section 3), we present a finite-sample probabilistic bound for $A_m$ that holds for any $m$ and provides the optimal rate of $m^{-1/2}$ for any compact set $\mathcal{S}$ along with guaranteeing the almost sure convergence of $A_m$ as long as the diameter of $\mathcal{S}$ is $e^{o(m)}$. Since convergence in uniform norm might sometimes be a too strong requirement and may not be suitable to attain correct rates in the generalization bounds associated with learning algorithms involving RFF,[2] we also study the behavior of $k(\mathbf{x}, \mathbf{y}) - \langle \phi(\mathbf{x}), \phi(\mathbf{y}) \rangle_{\mathbb{R}^{2m}}$ in $L^r$-norm ($1 \le r < \infty$) and obtain an optimal rate of $m^{-1/2}$. The RFF approach to approximate a translation-invariant kernel can be seen as a special of the problem of approximating a function in the barycenter of a family (say $\mathcal{F}$) of functions, which was considered in [14]. However, the approximation guarantees in [14, Theorem 3.2] do not directly apply to RFF as the assumptions on $\mathcal{F}$ are not satisfied by the cosine function, which is the family of functions that is used to approximate the kernel in the RFF approach. While a careful modification of the proof of [14, Theorem 3.2] could yield $m^{-1/2}$ rate of approximation for any compact set $\mathcal{S}$, this result would still be sub-optimal by providing a linear dependence on $|\mathcal{S}|$ similar to the theorems in [13, 22], in contrast to the optimal logarithmic dependence on $|\mathcal{S}|$ that is guaranteed by our results.

Traditionally, kernel based algorithms involve computing the value of the kernel. Recently, kernel algorithms involving the derivatives of the kernel (i.e., the Gram matrix consists of derivatives of the kernel computed at training samples) have been used to address numerous machine learning tasks, e.g., semi-supervised or Hermite learning with gradient information [28, 18], nonlinear variable selection [15, 16], (multi-task) gradient learning [26] and fitting of distributions in an infinite-dimensional exponential family [20]. Given the importance of these derivative based kernel algorithms, similar to [13], in Section 4, we propose a finite dimensional random feature map approximation to kernel derivatives, which can be used to speed up the above mentioned derivative based kernel algorithms. We present a finite-sample bound that quantifies the quality of approximation in uniform and $L^r$-norms and show the rate of convergence to be $m^{-1/2}$ in both these cases.

A summary of our *contributions* are as follows. We

1. provide the first detailed finite-sample performance analysis of RFFs for approximating kernels and their derivatives.
2. prove uniform and $L^r$ convergence on fixed compacts sets with optimal rate in terms of the RFF dimension ($m$);
3. give sufficient conditions for the growth rate of compact sets while preserving a.s. convergence uniformly and in $L^r$; specializing our result we match the best attainable asymptotic growth rate.

Various notations and definitions that are used throughout the paper are provided in Section 2 along with a brief review of RFF approximation proposed by [13]. The missing proofs of the results in Sections 3 and 4 are provided in the supplementary material.

## 2 Notations & preliminaries

In this section, we introduce notations that are used throughout the paper and then present preliminaries on kernel approximation through random feature maps as introduced by [13].

**Definitions & Notation:** For a topological space $\mathcal{X}$, $C(\mathcal{X})$ (*resp.* $C_b(\mathcal{X})$) denotes the space of all continuous (*resp.* bounded continuous) functions on $\mathcal{X}$. For $f \in C_b(\mathcal{X})$, $\|f\|_{\mathcal{X}} := \sup_{x \in \mathcal{X}} |f(x)|$ is the supremum norm of $f$. $M_b(\mathcal{X})$ and $M_+^1(\mathcal{X})$ is the set of all finite Borel and probability measures on $\mathcal{X}$, respectively. For $\mu \in M_b(\mathcal{X})$, $L^r(\mathcal{X}, \mu)$ denotes the Banach space of $r$-power ($r \geq 1$) $\mu$-integrable functions. For $\mathcal{X} \subseteq \mathbb{R}^d$, we will use $L^r(\mathcal{X})$ for $L^r(\mathcal{X}, \mu)$ if $\mu$ is a Lebesgue measure on $\mathcal{X}$. For $f \in L^r(\mathcal{X}, \mu)$, $\|f\|_{L^r(\mathcal{X}, \mu)} := \left( \int_{\mathcal{X}} |f|^r \, \mathrm{d}\mu \right)^{1/r}$ denotes the $L^r$-norm of $f$ for $1 \leq r < \infty$ and we write it as $\|\cdot\|_{L^r(\mathcal{X})}$ if $\mathcal{X} \subseteq \mathbb{R}^d$ and $\mu$ is the Lebesgue measure. For any $f \in L^1(\mathcal{X}, \mathbb{P})$ where $\mathbb{P} \in M_+^1(\mathcal{X})$, we define $\mathbb{P}f := \int_{\mathcal{X}} f(x) \, \mathrm{d}\mathbb{P}(x)$ and $\mathbb{P}_m f := \frac{1}{m} \sum_{i=1}^m f(X_i)$ where $(X_i)_{i=1}^m \overset{i.i.d.}{\sim} \mathbb{P}$, $\mathbb{P}_m := \frac{1}{m} \sum_{i=1}^m \delta_{X_i}$ is the empirical measure and $\delta_x$ is a Dirac measure supported on $x \in \mathcal{X}$. $\mathrm{supp}(\mathbb{P})$ denotes the support of $\mathbb{P}$. $\mathbb{P}^m := \otimes_{j=1}^m \mathbb{P}$ denotes the $m$-fold product measure.

For $\mathbf{v} := (v_1, \ldots, v_d) \in \mathbb{R}^d$, $\|\mathbf{v}\|_2 := \sqrt{\sum_{i=1}^d v_i^2}$. The diameter of $A \subseteq \mathcal{Y}$ where $(\mathcal{Y}, \rho)$ is a metric space is defined as $|A|_\rho := \sup\{\rho(x, y) : x, y \in \mathcal{Y}\}$. If $\mathcal{Y} = \mathbb{R}^d$ with $\rho = \|\cdot\|_2$, we denote the diameter of $A$ as $|A|$; $|A| < \infty$ if $A$ is compact. The volume of $A \subseteq \mathbb{R}^d$ is defined as $\mathrm{vol}(A) = \int_A 1 \, \mathrm{d}\mathbf{x}$. For $A \subseteq \mathbb{R}^d$, we define $A_\Delta := A - A = \{\mathbf{x} - \mathbf{y} : \mathbf{x}, \mathbf{y} \in A\}$. $conv(A)$ is the convex hull of $A$. For a function $g$ defined on open set $B \subseteq \mathbb{R}^d \times \mathbb{R}^d$, $\partial^{\mathbf{p}, \mathbf{q}} g(\mathbf{x}, \mathbf{y}) := \frac{\partial^{|\mathbf{p}| + |\mathbf{q}|} g(\mathbf{x}, \mathbf{y})}{\partial x_1^{p_1} \cdots \partial x_d^{p_d} \partial y_1^{q_1} \cdots \partial y_d^{q_d}}$, $(\mathbf{x}, \mathbf{y}) \in B$, where $\mathbf{p}, \mathbf{q} \in \mathbb{N}^d$ are multi-indices, $|\mathbf{p}| = \sum_{j=1}^d p_j$ and $\mathbb{N} := \{0, 1, 2, \ldots\}$. Define $\mathbf{v}^{\mathbf{p}} = \prod_{j=1}^d v_j^{p_j}$. For positive sequences $(a_n)_{n \in \mathbb{N}}$, $(b_n)_{n \in \mathbb{N}}$, $a_n = o(b_n)$ if $\lim_{n \to \infty} \frac{a_n}{b_n} = 0$. $X_n = O_p(r_n)$ (*resp.* $O_{a.s.}(r_n)$) denotes that $\frac{X_n}{r_n}$ is bounded in probability (*resp.* almost surely). $\Gamma(t) = \int_0^\infty x^{t-1} e^{-x} \, \mathrm{d}x$ is the Gamma function, $\Gamma\left(\frac{1}{2}\right) = \sqrt{\pi}$ and $\Gamma(t+1) = t\Gamma(t)$.

**Random feature maps:** Let $k : \mathbb{R}^d \times \mathbb{R}^d \to \mathbb{R}$ be a bounded, continuous, positive definite, translation-invariant kernel, i.e., there exists a positive definite function $\psi : \mathbb{R}^d \to \mathbb{R}$ such that $k(\mathbf{x}, \mathbf{y}) = \psi(\mathbf{x} - \mathbf{y})$, $\mathbf{x}, \mathbf{y} \in \mathbb{R}^d$ where $\psi \in C_b(\mathbb{R}^d)$. By Bochner's theorem [24, Theorem 6.6], $\psi$ can be represented as the Fourier transform of a finite non-negative Borel measure $\Lambda$ on $\mathbb{R}^d$, i.e.,

$$k(\mathbf{x}, \mathbf{y}) = \psi(\mathbf{x} - \mathbf{y}) = \int_{\mathbb{R}^d} e^{\sqrt{-1} \boldsymbol{\omega}^T (\mathbf{x} - \mathbf{y})} \mathrm{d}\Lambda(\boldsymbol{\omega}) \overset{(\star)}{=} \int_{\mathbb{R}^d} \cos\left(\boldsymbol{\omega}^T (\mathbf{x} - \mathbf{y})\right) \mathrm{d}\Lambda(\boldsymbol{\omega}), \qquad (1)$$

where $(\star)$ follows from the fact that $\psi$ is real-valued and symmetric. Since $\Lambda(\mathbb{R}^d) = \psi(0)$, $k(\mathbf{x}, \mathbf{y}) = \psi(0) \int e^{\sqrt{-1} \boldsymbol{\omega}^T (\mathbf{x} - \mathbf{y})} \, \mathrm{d}\mathbb{P}(\boldsymbol{\omega})$ where $\mathbb{P} := \frac{\Lambda}{\psi(0)} \in M_+^1(\mathbb{R}^d)$. Therefore, w.l.o.g., we assume throughout the paper that $\psi(0) = 1$ and so $\Lambda \in M_+^1(\mathbb{R}^d)$. Based on (1), [13] proposed an approximation to $k$ by replacing $\Lambda$ with its empirical measure, $\Lambda_m$ constructed from $(\boldsymbol{\omega}_i)_{i=1}^m \overset{i.i.d.}{\sim} \Lambda$ so that resultant approximation can be written as the Euclidean inner product of finite dimensional random feature maps, i.e.,

$$\hat{k}(\mathbf{x}, \mathbf{y}) = \frac{1}{m} \sum_{i=1}^m \cos\left(\boldsymbol{\omega}_i^T (\mathbf{x} - \mathbf{y})\right) \overset{(*)}{=} \langle \phi(\mathbf{x}), \phi(\mathbf{y}) \rangle_{\mathbb{R}^{2m}}, \qquad (2)$$

where $\phi(\mathbf{x}) = \frac{1}{\sqrt{m}} (\cos(\boldsymbol{\omega}_1^T \mathbf{x}), \ldots, \cos(\boldsymbol{\omega}_m^T \mathbf{x}), \sin(\boldsymbol{\omega}_1^T \mathbf{x}), \ldots, \sin(\boldsymbol{\omega}_m^T \mathbf{x}))$ and $(*)$ holds based on the basic trigonometric identity: $\cos(a - b) = \cos a \cos b + \sin a \sin b$. This elegant approximation to $k$ is particularly useful in speeding up kernel-based algorithms as the finite-dimensional random feature map $\phi$ can be used to solve these algorithms in the primal thereby offering better computational complexity (than by solving them in the dual) while at the same time not lacking in performance. Apart from these practical advantages, [13, Claim 1] (and similarly, [22, Prop. 1]) provides a theoretical guarantee that $\|\hat{k} - k\|_{\mathcal{S} \times \mathcal{S}} \to 0$ as $m \to \infty$ for any compact set $\mathcal{S} \subset \mathbb{R}^d$. Formally, [13, Claim

1] showed that—note that (3) is slightly different but more precise than the one in the statement of Claim 1 in [13]—for any $\epsilon > 0$,

$$\Lambda^m \left( \left\{ (\boldsymbol{\omega}_i)_{i=1}^m : \|\hat{k} - k\|_{\mathcal{S} \times \mathcal{S}} \geq \epsilon \right\} \right) \leq C_d \left( |\mathcal{S}| \sigma \epsilon^{-1} \right)^{\frac{2d}{d+2}} e^{-\frac{m\epsilon^2}{4(d+2)}}, \qquad (3)$$

where $\sigma^2 := \int \|\boldsymbol{\omega}\|^2 \, d\Lambda(\boldsymbol{\omega})$ and $C_d := 2^{\frac{6d+2}{d+2}} \left( \left(\frac{2}{d}\right)^{\frac{d}{d+2}} + \left(\frac{d}{2}\right)^{\frac{2}{d+2}} \right) \leq 2^7 d^{\frac{2}{d+2}}$ when $d \geq 2$. The condition $\sigma^2 < \infty$ implies that $\psi$ (and therefore $k$) is twice differentiable. From (3) it is clear that the probability has polynomial tails if $\epsilon < |\mathcal{S}|\sigma$ (i.e., small $\epsilon$) and Gaussian tails if $\epsilon \geq |\mathcal{S}|\sigma$ (i.e., large $\epsilon$) and can be equivalently written as

$$\Lambda^m \left( \left\{ (\boldsymbol{\omega}_i)_{i=1}^m : \|\hat{k} - k\|_{\mathcal{S} \times \mathcal{S}} \geq C_d^{\frac{d+2}{2d}} |\mathcal{S}| \sigma \sqrt{m^{-1} \log m} \right\} \right) \leq m^{\frac{\alpha}{4(d+2)}} (\log m)^{-\frac{d}{d+2}}, \qquad (4)$$

where $\alpha := 4d - C_d^{\frac{d+2}{d}} |\mathcal{S}|^2 \sigma^2$. For $|\mathcal{S}|$ sufficiently large (i.e., $\alpha < 0$), it follows from (4) that

$$\|\hat{k} - k\|_{\mathcal{S} \times \mathcal{S}} = O_p \left( |\mathcal{S}| \sqrt{m^{-1} \log m} \right). \qquad (5)$$

While (5) shows that $\hat{k}$ is a consistent estimator of $k$ in the topology of compact convergence (i.e., $\hat{k}$ convergences to $k$ uniformly over compact sets), the rate of convergence of $\sqrt{(\log m)/m}$ is not optimal. In addition, the order of dependence on $|\mathcal{S}|$ is not optimal. While a faster rate (in fact, an optimal rate) of convergence is desired—better rates in (5) can lead to better convergence rates for the excess error of the kernel machine constructed using $\hat{k}$—, the order of dependence on $|\mathcal{S}|$ is also important as it determines the the number of RFF features (i.e., $m$) that are needed to achieve a given approximation accuracy. In fact, the order of dependence on $|\mathcal{S}|$ controls the rate at which $|\mathcal{S}|$ can be grown as a function of $m$ when $m \to \infty$ (see Remark 1*(ii)* for a detailed discussion about the significance of growing $|\mathcal{S}|$). In the following section, we present an analogue of (4)—see Theorem 1—that provides optimal rates and has correct dependence on $|\mathcal{S}|$.

## 3    Main results: approximation of $k$

As discussed in Sections 1 and 2, while the random feature map approximation of $k$ introduced by [13] has many practical advantages, it does not seem to be theoretically well-understood. The existing theoretical results on the quality of approximation do not provide a complete picture owing to their non-optimality. In this section, we first present our main result (see Theorem 1) that improves upon (4) and provides a rate of $m^{-1/2}$ with logarithm dependence on $|\mathcal{S}|$. We then discuss the consequences of Theorem 1 along with its optimality in Remark 1. Next, in Corollary 2 and Theorem 3, we discuss the $L^r$-convergence ($1 \leq r < \infty$) of $\hat{k}$ to $k$ over compact subsets of $\mathbb{R}^d$.

**Theorem 1.** *Suppose $k(\mathbf{x}, \mathbf{y}) = \psi(\mathbf{x} - \mathbf{y})$, $\mathbf{x}, \mathbf{y} \in \mathbb{R}^d$ where $\psi \in C_b(\mathbb{R}^d)$ is positive definite and $\sigma^2 := \int \|\boldsymbol{\omega}\|^2 \, \mathrm{d}\Lambda(\boldsymbol{\omega}) < \infty$. Then for any $\tau > 0$ and non-empty compact set $\mathcal{S} \subset \mathbb{R}^d$,*

$$\Lambda^m \left( \left\{ (\boldsymbol{\omega}_i)_{i=1}^m : \|\hat{k} - k\|_{\mathcal{S} \times \mathcal{S}} \geq \frac{h(d, |\mathcal{S}|, \sigma) + \sqrt{2\tau}}{\sqrt{m}} \right\} \right) \leq e^{-\tau},$$

*where $h(d, |\mathcal{S}|, \sigma) := 32\sqrt{2d \log(2|\mathcal{S}| + 1)} + 32\sqrt{2d \log(\sigma + 1)} + 16\sqrt{2d[\log(2|\mathcal{S}| + 1)]^{-1}}$.*

*Proof (sketch).* Note that $\|\hat{k} - k\|_{\mathcal{S} \times \mathcal{S}} = \sup_{\mathbf{x}, \mathbf{y} \in \mathcal{S}} |\hat{k}(\mathbf{x}, \mathbf{y}) - k(\mathbf{x}, \mathbf{y})| = \sup_{g \in \mathcal{G}} |\Lambda_m g - \Lambda g|$, where $\mathcal{G} := \{g_{\mathbf{x}, \mathbf{y}}(\boldsymbol{\omega}) = \cos(\boldsymbol{\omega}^T(\mathbf{x} - \mathbf{y})) : \mathbf{x}, \mathbf{y} \in \mathcal{S}\}$, which means the object of interest is the suprema of an empirical process indexed by $\mathcal{G}$. Instead of bounding $\sup_{g \in \mathcal{G}} |\Lambda_m g - \Lambda g|$ by using Hoeffding's inequality on a cover of $\mathcal{G}$ and then applying union bound as carried out in [13, 22], we use the refined technique of applying concentration via McDiarmid's inequality, followed by symmetrization and bound the Rademacher average by Dudley entropy bound. The result is obtained by carefully bounding the $L^2(\Lambda_m)$-covering number of $\mathcal{G}$. The details are provided in Section B.1 of the supplementary material. $\qquad \square$

*Remark* 1. *(i)* Theorem 1 shows that $\hat{k}$ is a consistent estimator of $k$ in the topology of compact convergence as $m \to \infty$ with the rate of a.s. convergence being $\sqrt{m^{-1} \log |\mathcal{S}|}$ (almost sure convergence is guaranteed by the first Borel-Cantelli lemma). In comparison to (4), it is clear that Theorem 1

provides improved rates with better constants and logarithmic dependence on $|\mathcal{S}|$ instead of a linear dependence. The logarithmic dependence on $|\mathcal{S}|$ ensures that we need $m = O(\epsilon^{-2} \log |\mathcal{S}|)$ random features instead of $O(\epsilon^{-2}|\mathcal{S}|^2 \log(|\mathcal{S}|/\epsilon))$ random features, i.e., significantly fewer features to achieve the same approximation accuracy of $\epsilon$.

*(ii)* **Growing diameter:** While Theorem 1 provides almost sure convergence uniformly over compact sets, one might wonder whether it is possible to achieve uniform convergence over $\mathbb{R}^d$. [7, Section 2] showed that such a result is possible if $\Lambda$ is a discrete measure but not possible for $\Lambda$ that is absolutely continuous w.r.t. the Lebesgue measure (i.e., if $\Lambda$ has a density). Since uniform convergence of $\hat{k}$ to $k$ over $\mathbb{R}^d$ is not possible for many interesting $k$ (e.g., Gaussian kernel), it is of interest to study the convergence on $\mathcal{S}$ whose diameter grows with $m$. Therefore, as mentioned in Section 2, the order of dependence of rates on $|\mathcal{S}|$ is critical. Suppose $|\mathcal{S}_m| \to \infty$ as $m \to \infty$ (we write $|\mathcal{S}_m|$ instead of $|\mathcal{S}|$ to show the explicit dependence on $m$). Then Theorem 1 shows that $\hat{k}$ is a consistent estimator of $k$ in the topology of compact convergence if $m^{-1} \log |\mathcal{S}_m| \to 0$ as $m \to \infty$ (i.e., $|\mathcal{S}_m| = e^{o(m)}$) in contrast to the result in (4) which requires $|\mathcal{S}_m| = o(\sqrt{m/\log m})$. In other words, Theorem 1 ensures consistency even when $|\mathcal{S}_m|$ grows exponentially in $m$ whereas (4) ensures consistency only if $|\mathcal{S}_m|$ does not grow faster than $\sqrt{m/\log m}$.

*(iii)* **Optimality:** Note that $\psi$ is the characteristic function of $\Lambda \in M_+^1(\mathbb{R}^d)$ since $\psi$ is the Fourier transform of $\Lambda$ (by Bochner's theorem). Therefore, the object of interest $\|\hat{k} - k\|_{\mathcal{S} \times \mathcal{S}} = \|\hat{\psi} - \psi\|_{\mathcal{S}_\Delta}$, is the uniform norm of the difference between $\psi$ and the empirical characteristic function $\hat{\psi} = \frac{1}{m} \sum_{i=1}^{m} \cos(\langle \boldsymbol{\omega}_i, \cdot \rangle)$, when both are restricted to a compact set $\mathcal{S}_\Delta \subset \mathbb{R}^d$. The question of the convergence behavior of $\|\hat{\psi} - \psi\|_{\mathcal{S}_\Delta}$ is not new and has been studied in great detail in the probability and statistics literature (e.g., see [7, 27] for $d = 1$ and [4, 5] for $d > 1$) where the characteristic function is not just a real-valued symmetric function (like $\psi$) but is Hermitian. [27, Theorems 6.2 and 6.3] show that the optimal rate of convergence of $\|\hat{\psi} - \psi\|_{\mathcal{S}_\Delta}$ is $m^{-1/2}$ when $d = 1$, which matches with our result in Theorem 1. Also Theorems 1 and 2 in [5] show that the logarithmic dependence on $|\mathcal{S}_m|$ is optimal asymptotically. In particular, [5, Theorem 1] matches with the growing diameter result in Remark 1*(ii)*, while [5, Theorem 2] shows that if $\Lambda$ is absolutely continuous w.r.t. the Lebesgue measure and if $\limsup_{m \to \infty} m^{-1} \log |\mathcal{S}_m| > 0$, then there exists a positive $\varepsilon$ such that $\limsup_{m \to \infty} \Lambda^m(\|\hat{\psi} - \psi\|_{\mathcal{S}_{m,\Delta}} \geq \varepsilon) > 0$. This means the rate $|\mathcal{S}_m| = e^{o(m)}$ is not only the best possible in general for almost sure convergence, but if faster sequence $|\mathcal{S}_m|$ is considered then even stochastic convergence cannot be retained for any characteristic function vanishing at infinity along at least one path. While these previous results match with that of Theorem 1 (and its consequences), we would like to highlight the fact that all these previous results are asymptotic in nature whereas Theorem 1 provides a finite-sample probabilistic inequality that holds for any $m$. We are not aware of any such finite-sample result except for the one in [13, 22]. ∎

Using Theorem 1, one can obtain a probabilistic inequality for the $L^r$-norm of $\hat{k} - k$ over any compact set $\mathcal{S} \subset \mathbb{R}^d$, as given by the following result.

**Corollary 2.** *Suppose $k$ satisfies the assumptions in Theorem 1. Then for any $1 \leq r < \infty$, $\tau > 0$ and non-empty compact set $\mathcal{S} \subset \mathbb{R}^d$,*

$$\Lambda^m \left( \left\{ (\boldsymbol{\omega}_i)_{i=1}^m : \|\hat{k} - k\|_{L^r(\mathcal{S})} \geq \left( \frac{\pi^{d/2}|\mathcal{S}|^d}{2^d \Gamma(\frac{d}{2} + 1)} \right)^{2/r} \frac{h(d, |\mathcal{S}|, \sigma) + \sqrt{2\tau}}{\sqrt{m}} \right\} \right) \leq e^{-\tau},$$

*where $\|\hat{k} - k\|_{L^r(\mathcal{S})} := \|\hat{k} - k\|_{L^r(\mathcal{S} \times \mathcal{S})} = \left( \int_{\mathcal{S}} \int_{\mathcal{S}} |\hat{k}(\mathbf{x}, \mathbf{y}) - k(\mathbf{x}, \mathbf{y})|^r \, d\mathbf{x} \, d\mathbf{y} \right)^{\frac{1}{r}}$.*

*Proof.* Note that
$$\|\hat{k} - k\|_{L^r(\mathcal{S})} \leq \|\hat{k} - k\|_{\mathcal{S} \times \mathcal{S}} \text{vol}^{2/r}(\mathcal{S}).$$
The result follows by combining Theorem 1 and the fact that $\text{vol}(\mathcal{S}) \leq \text{vol}(A)$ where $A := \left\{ \mathbf{x} \in \mathbb{R}^d : \|\mathbf{x}\|_2 \leq \frac{|\mathcal{S}|}{2} \right\}$ and $\text{vol}(A) = \frac{\pi^{d/2}|\mathcal{S}|^d}{2^d \Gamma(\frac{d}{2} + 1)}$ (which follows from [8, Corollary 2.55]). □

Corollary 2 shows that $\|\hat{k} - k\|_{L^r(\mathcal{S})} = O_{a.s.}(m^{-1/2}|\mathcal{S}|^{2d/r} \sqrt{\log |\mathcal{S}|})$ and therefore if $|\mathcal{S}_m| \to \infty$ as $m \to \infty$, then consistency of $\hat{k}$ in $L^r(\mathcal{S}_m)$-norm is achieved as long as $m^{-1/2}|\mathcal{S}_m|^{2d/r} \sqrt{\log |\mathcal{S}_m|} \to$

0 as $m \to \infty$. This means, in comparison to the uniform norm in Theorem 1 where $|\mathcal{S}_m|$ can grow exponential in $m^\delta$ ($\delta < 1$), $|\mathcal{S}_m|$ cannot grow faster than $m^{\frac{r}{4d}} (\log m)^{-\frac{r}{4d} - \theta}$ ($\theta > 0$) to achieve consistency in $L^r$-norm.

Instead of using Theorem 1 to obtain a bound on $\|\hat{k} - k\|_{L^r(\mathcal{S})}$ (this bound may be weak as $\|\hat{k} - k\|_{L^r(\mathcal{S})} \le \|\hat{k} - k\|_{\mathcal{S} \times \mathcal{S}} \mathrm{vol}^{2/r}(\mathcal{S})$ for any $1 \le r < \infty$), a better bound (for $2 \le r < \infty$) can be obtained by directly bounding $\|\hat{k} - k\|_{L^r(\mathcal{S})}$, as shown in the following result.

**Theorem 3.** *Suppose $k(\mathbf{x}, \mathbf{y}) = \psi(\mathbf{x} - \mathbf{y})$, $\mathbf{x}$, $\mathbf{y} \in \mathbb{R}^d$ where $\psi \in C_b(\mathbb{R}^d)$ is positive definite. Then for any $1 < r < \infty$, $\tau > 0$ and non-empty compact set $\mathcal{S} \subset \mathbb{R}^d$,*

$$\Lambda^m \left( \left\{ (\boldsymbol{\omega}_i)_{i=1}^m : \|\hat{k} - k\|_{L^r(\mathcal{S})} \ge \left( \frac{\pi^{d/2} |\mathcal{S}|^d}{2^d \Gamma(\frac{d}{2} + 1)} \right)^{2/r} \left( \frac{C_r'}{m^{1 - \max\{\frac{1}{2}, \frac{1}{r}\}}} + \frac{\sqrt{2\tau}}{\sqrt{m}} \right) \right\} \right) \le e^{-\tau},$$

*where $C_r'$ is the Khintchine constant given by $C_r' = 1$ for $r \in (1, 2]$ and $C_r' = \sqrt{2} \left[ \Gamma\left(\frac{r+1}{2}\right) / \sqrt{\pi} \right]^{\frac{1}{r}}$ for $r \in [2, \infty)$.*

*Proof (sketch).* As in Theorem 1, we show that $\|k - \hat{k}\|_{L^r(\mathcal{S})}$ satisfies the bounded difference property, hence by the McDiarmid's inequality, it concentrates around its expectation $\mathbb{E}\|k - \hat{k}\|_{L^r(\mathcal{S})}$. By symmetrization, we then show that $\mathbb{E}\|k - \hat{k}\|_{L^r(\mathcal{S})}$ is upper bounded in terms of $\mathbb{E}_{\boldsymbol{\varepsilon}} \|\sum_{i=1}^m \varepsilon_i \cos(\langle \boldsymbol{\omega}_i, \cdot - \cdot\rangle)\|_{L^r(\mathcal{S})}$, where $\boldsymbol{\varepsilon} := (\varepsilon_i)_{i=1}^m$ are Rademacher random variables. By exploiting the fact that $L^r(\mathcal{S})$ is a Banach space of type $\min\{r, 2\}$, the result follows. The details are provided in Section B.2 of the supplementary material. $\square$

*Remark* 2. Theorem 3 shows an improved dependence on $|\mathcal{S}|$ without the extra $\sqrt{\log |\mathcal{S}|}$ factor given in Corollary 2 and therefore provides a better rate for $2 \le r < \infty$ when the diameter of $\mathcal{S}$ grows, i.e., $\|\hat{k} - k\|_{L^r(\mathcal{S}_m)} \overset{a.s.}{\to} 0$ if $|\mathcal{S}_m| = o(m^{\frac{r}{4d}})$ as $m \to \infty$. However, for $1 < r < 2$, Theorem 3 provides a slower rate than Corollary 2 and therefore it is appropriate to use the bound in Corollary 2. While one might wonder why we only considered the convergence of $\|\hat{k} - k\|_{L^r(\mathcal{S})}$ and not $\|\hat{k} - k\|_{L^r(\mathbb{R}^d)}$, it is important to note that the latter is not well-defined because $\hat{k} \notin L^r(\mathbb{R}^d)$ even if $k \in L^r(\mathbb{R}^d)$. $\blacksquare$

# 4 Approximation of kernel derivatives

In the previous section we focused on the approximation of the kernel function where we presented uniform and $L^r$ convergence guarantees on compact sets for the random Fourier feature approximation, and discussed how fast the diameter of these sets can grow to preserve uniform and $L^r$ convergence almost surely. In this section, we propose an approximation to derivatives of the kernel and analyze the uniform and $L^r$ convergence behavior of the proposed approximation. As motivated in Section 1, the question of approximating the derivatives of the kernel through finite dimensional random feature map is also important as it enables to speed up several interesting machine learning tasks that involve the derivatives of the kernel [28, 18, 15, 16, 26, 20], see for example the recent infinite dimensional exponential family fitting technique [21], which implements this idea.

To this end, we consider $k$ as in (1) and define $h_a := \cos(\frac{\pi a}{2} + \cdot)$, $a \in \mathbb{N}$ (in other words $h_0 = \cos$, $h_1 = -\sin$, $h_2 = -\cos$, $h_3 = \sin$ and $h_a = h_{a \bmod 4}$). For $\mathbf{p}, \mathbf{q} \in \mathbb{N}^d$, assuming $\int |\boldsymbol{\omega}^{\mathbf{p}+\mathbf{q}}| \, d\Lambda(\boldsymbol{\omega}) < \infty$, it follows from the dominated convergence theorem that

$$\partial^{\mathbf{p}, \mathbf{q}} k(\mathbf{x}, \mathbf{y}) = \int_{\mathbb{R}^d} \boldsymbol{\omega}^{\mathbf{p}} (-\boldsymbol{\omega})^{\mathbf{q}} h_{|\mathbf{p}+\mathbf{q}|} \left( \boldsymbol{\omega}^T (\mathbf{x} - \mathbf{y}) \right) d\Lambda(\boldsymbol{\omega})$$

$$= \int_{\mathbb{R}^d} \boldsymbol{\omega}^{\mathbf{p}+\mathbf{q}} \left[ h_{|\mathbf{p}|}(\boldsymbol{\omega}^T \mathbf{x}) h_{|\mathbf{q}|}(\boldsymbol{\omega}^T \mathbf{y}) + h_{3+|\mathbf{p}|}(\boldsymbol{\omega}^T \mathbf{x}) h_{3+|\mathbf{q}|}(\boldsymbol{\omega}^T \mathbf{y}) \right] d\Lambda(\boldsymbol{\omega}),$$

so that $\partial^{\mathbf{p}, \mathbf{q}} k(\mathbf{x}, \mathbf{y})$ can be approximated by replacing $\Lambda$ with $\Lambda_m$, resulting in

$$\widehat{\partial^{\mathbf{p}, \mathbf{q}} k}(\mathbf{x}, \mathbf{y}) := s^{\mathbf{p}, \mathbf{q}}(\mathbf{x}, \mathbf{y}) = \frac{1}{m} \sum_{j=1}^m \boldsymbol{\omega}_j^{\mathbf{p}} (-\boldsymbol{\omega}_j)^{\mathbf{q}} h_{|\mathbf{p}+\mathbf{q}|} \left( \boldsymbol{\omega}_j^T (\mathbf{x} - \mathbf{y}) \right) = \langle \phi^{\mathbf{p}}(\mathbf{x}), \phi^{\mathbf{q}}(\mathbf{y}) \rangle_{\mathbb{R}^{2m}}, \quad (6)$$

where $\phi^{\mathbf{P}}(\mathbf{u}) := \frac{1}{\sqrt{m}} \left( \boldsymbol{\omega}_1^{\mathbf{P}} h_{|\mathbf{p}|}(\boldsymbol{\omega}_1^T \mathbf{u}), \cdots, \boldsymbol{\omega}_m^{\mathbf{P}} h_{|\mathbf{p}|}(\boldsymbol{\omega}_m^T \mathbf{u}), \boldsymbol{\omega}_1^{\mathbf{P}} h_{3+|\mathbf{p}|}(\boldsymbol{\omega}_1^T \mathbf{u}), \cdots, \boldsymbol{\omega}_m^{\mathbf{P}} h_{3+|\mathbf{p}|}(\boldsymbol{\omega}_m^T \mathbf{u}) \right)$
and $(\boldsymbol{\omega}_j)_{j=1}^m \overset{i.i.d.}{\sim} \Lambda$. Now the goal is to understand the behavior of $\|s^{\mathbf{p},\mathbf{q}} - \partial^{\mathbf{p},\mathbf{q}}k\|_{\mathbb{S} \times \mathbb{S}}$ and $\|s^{\mathbf{p},\mathbf{q}} - \partial^{\mathbf{p},\mathbf{q}}k\|_{L^r(\mathbb{S})}$ for $r \in [1, \infty)$, i.e., obtain analogues of Theorems 1 and 3.

As in the proof sketch of Theorem 1, while $\|s^{\mathbf{p},\mathbf{q}} - \partial^{\mathbf{p},\mathbf{q}}k\|_{\mathbb{S} \times \mathbb{S}}$ can be analyzed as the suprema of an empirical process indexed by a suitable function class (say $\mathcal{G}$), some technical issues arise because $\mathcal{G}$ is not uniformly bounded. This means McDiarmid or Talagrand's inequality cannot be applied to achieve concentration and bounding Rademacher average by Dudley entropy bound may not be reasonable. While these issues can be tackled by resorting to more technical and refined methods, in this paper, we generalize (see Theorem 4 which is proved in Section B.1 of the supplement) Theorem 1 to derivatives under the restrictive assumption that $\mathrm{supp}(\Lambda)$ is bounded (note that many popular kernels including the Gaussian do not satisfy this assumption). We also present another result (see Theorem 5) by generalizing the proof technique[3] of [13] to *unbounded* functions where the boundedness assumption of $\mathrm{supp}(\Lambda)$ is relaxed but at the expense of a worse rate (compared to Theorem 4).

**Theorem 4.** *Let* $\mathbf{p}, \mathbf{q} \in \mathbb{N}^d$, $T_{\mathbf{p},\mathbf{q}} := \sup_{\boldsymbol{\omega} \in \mathrm{supp}(\Lambda)} |\boldsymbol{\omega}^{\mathbf{p}+\mathbf{q}}|$, $C_{\mathbf{p},\mathbf{q}} := \mathbb{E}_{\boldsymbol{\omega} \sim \Lambda} \left[ |\boldsymbol{\omega}^{\mathbf{p}+\mathbf{q}}| \, \|\boldsymbol{\omega}\|_2^2 \right]$, *and assume that* $C_{2\mathbf{p},2\mathbf{q}} < \infty$. *Suppose* $\mathrm{supp}(\Lambda)$ *is bounded if* $\mathbf{p} \neq \mathbf{0}$ *and* $\mathbf{q} \neq \mathbf{0}$. *Then for any* $\tau > 0$ *and non-empty compact set* $\mathbb{S} \subset \mathbb{R}^d$,

$$\Lambda^m \left( \left\{ (\boldsymbol{\omega}_i)_{i=1}^m : \|\partial^{\mathbf{p},\mathbf{q}}k - s^{\mathbf{p},\mathbf{q}}\|_{\mathbb{S} \times \mathbb{S}} \geq \frac{H(d, \mathbf{p}, \mathbf{q}, |\mathbb{S}|) + T_{\mathbf{p},\mathbf{q}}\sqrt{2\tau}}{\sqrt{m}} \right\} \right) \leq e^{-\tau},$$

*where*

$$H(d, \mathbf{p}, \mathbf{q}, |\mathbb{S}|) = 32\sqrt{2d\,T_{2\mathbf{p},2\mathbf{q}}} \left[ \sqrt{U(\mathbf{p}, \mathbf{q}, |\mathbb{S}|)} + \frac{1}{2\sqrt{U(\mathbf{p}, \mathbf{q}, |\mathbb{S}|)}} + \sqrt{\log(\sqrt{C_{2\mathbf{p},2\mathbf{q}}} + 1)} \right],$$

$$U(\mathbf{p}, \mathbf{q}, |\mathbb{S}|) = \log \left( 2|\mathbb{S}|T_{2\mathbf{p},2\mathbf{q}}^{-1/2} + 1 \right).$$

*Remark* 3. *(i)* Note that Theorem 4 reduces to Theorem 1 if $\mathbf{p} = \mathbf{q} = 0$, in which case $T_{\mathbf{p},\mathbf{q}} = T_{2\mathbf{p},2\mathbf{q}} = 1$. If $\mathbf{p} \neq \mathbf{0}$ or $\mathbf{q} \neq \mathbf{0}$, then the boundedness of $\mathrm{supp}(\Lambda)$ implies that $T_{\mathbf{p},\mathbf{q}} < \infty$ and $T_{2\mathbf{p},2\mathbf{q}} < \infty$.

*(ii)* **Growth of** $|\mathbb{S}_m|$**:** By the same reasoning as in Remark 1*(ii)* and Corollary 2, it follows that $\|\partial^{\mathbf{p},\mathbf{q}}k - s^{\mathbf{p},\mathbf{q}}\|_{\mathbb{S}_m \times \mathbb{S}_m} \xrightarrow{a.s.} 0$ if $|\mathbb{S}_m| = e^{o(m)}$ and $\|\partial^{\mathbf{p},\mathbf{q}}k - s^{\mathbf{p},\mathbf{q}}\|_{L^r(\mathbb{S}_m)} \xrightarrow{a.s.} 0$ if $m^{-1/2}|\mathbb{S}_m|^{2d/r}\sqrt{\log|\mathbb{S}_m|} \to 0$ (for $1 \leq r < \infty$) as $m \to \infty$. An exact analogue of Theorem 3 can be obtained (but with different constants) under the assumption that $\mathrm{supp}(\Lambda)$ is bounded and it can be shown that for $r \in [2, \infty)$, $\|\partial^{\mathbf{p},\mathbf{q}}k - s^{\mathbf{p},\mathbf{q}}\|_{L^r(\mathbb{S}_m)} \xrightarrow{a.s.} 0$ if $|\mathbb{S}_m| = o(m^{\frac{r}{4d}})$. ∎

The following result relaxes the boundedness of $\mathrm{supp}(\Lambda)$ by imposing certain moment conditions on $\Lambda$ but at the expense of a worse rate. The proof relies on applying Bernstein inequality at the elements of a net (which exists by the compactness of $\mathbb{S}$) combined with a union bound, and extending the approximation error from the anchors by a probabilistic Lipschitz argument.

**Theorem 5.** *Let* $\mathbf{p}, \mathbf{q} \in \mathbb{N}^d$, $\psi$ *be continuously differentiable,* $\mathbf{z} \mapsto \nabla_{\mathbf{z}}[\partial^{\mathbf{p},\mathbf{q}}k(\mathbf{z})]$ *be continuous,* $\mathbb{S} \subset \mathbb{R}^d$ *be any non-empty compact set,* $D_{\mathbf{p},\mathbf{q},\mathbb{S}} := \sup_{\mathbf{z} \in conv(\mathbb{S}_\Delta)} \|\nabla_{\mathbf{z}}[\partial^{\mathbf{p},\mathbf{q}}k(\mathbf{z})]\|_2$ *and* $E_{\mathbf{p},\mathbf{q}} := \mathbb{E}_{\boldsymbol{\omega} \sim \Lambda}[|\boldsymbol{\omega}^{\mathbf{p}+\mathbf{q}}| \, \|\boldsymbol{\omega}\|_2]$. *Assume that* $E_{\mathbf{p},\mathbf{q}} < \infty$. *Suppose* $\exists L > 0, \sigma > 0$ *such that*

$$\mathbb{E}_{\boldsymbol{\omega} \sim \Lambda} \left[ |f(\mathbf{z}; \boldsymbol{\omega})|^M \right] \leq \frac{M!\,\sigma^2 L^{M-2}}{2} \quad (\forall M \geq 2, \forall \mathbf{z} \in \mathbb{S}_\Delta), \tag{7}$$

*where* $f(\mathbf{z};\boldsymbol{\omega}) = \partial^{\mathbf{p},\mathbf{q}} k(\mathbf{z}) - \boldsymbol{\omega}^{\mathbf{p}}(-\boldsymbol{\omega})^{\mathbf{q}} h_{|\mathbf{p}+\mathbf{q}|}\left(\boldsymbol{\omega}^T \mathbf{z}\right)$. *Define* $F_d := d^{-\frac{d}{d+1}} + d^{\frac{1}{d+1}}$.[4] *Then*

$$\Lambda^m\left(\{(\boldsymbol{\omega}_i)_{i=1}^m : \|\partial^{\mathbf{p},\mathbf{q}} k - s^{\mathbf{p},\mathbf{q}}\|_{\mathcal{S}\times\mathcal{S}} \geq \epsilon\}\right) \leq$$

$$\leq 2^{d-1} e^{-\frac{m\epsilon^2}{8\sigma^2\left(1+\frac{\epsilon L}{2\sigma^2}\right)}} + F_d 2^{\frac{4d-1}{d+1}}\left[\frac{|\mathcal{S}|(D_{\mathbf{p},\mathbf{q},\mathcal{S}} + E_{\mathbf{p},\mathbf{q}})}{\epsilon}\right]^{\frac{d}{d+1}} e^{-\frac{m\epsilon^2}{8(d+1)\sigma^2\left(1+\frac{\epsilon L}{2\sigma^2}\right)}}. \quad (8)$$

*Remark* 4. *(i)* The compactness of $\mathcal{S}$ implies that of $\mathcal{S}_\Delta$. Hence, by the continuity of $\mathbf{z} \mapsto \nabla_{\mathbf{z}}\left[\partial^{\mathbf{p},\mathbf{q}} k(\mathbf{z})\right]$, one gets $D_{\mathbf{p},\mathbf{q},\mathcal{S}} < \infty$. (7) holds if $|f(\mathbf{z};\boldsymbol{\omega})| \leq \frac{L}{2}$ and $\mathbb{E}_{\boldsymbol{\omega}\sim\Lambda}\left[|f(\mathbf{z};\boldsymbol{\omega})|^2\right] \leq \sigma^2$ $(\forall \mathbf{z} \in \mathcal{S}_\Delta)$. If $\text{supp}(\Lambda)$ is bounded, then the boundedness of $f$ is guaranteed (see Section B.4 in the supplement).

*(ii)* In the special case when $\mathbf{p} = \mathbf{q} = \mathbf{0}$, our requirement boils down to the continuously differentiability of $\psi$, $E_{\mathbf{0},\mathbf{0}} = \mathbb{E}_{\boldsymbol{\omega}\sim\Lambda}\|\boldsymbol{\omega}\|_2 < \infty$, and (7).

*(iii)* Note that (8) is similar to (3) and therefore based on the discussion in Section 2, one has $\|\partial^{\mathbf{p},\mathbf{q}} k - s^{\mathbf{p},\mathbf{q}}\|_{\mathcal{S}\times\mathcal{S}} = O_{a.s.}(|\mathcal{S}|\sqrt{m^{-1}\log m})$. But the advantage with Theorem 5 over [13, Claim 1] and [22, Prop. 1] is that it can handle unbounded functions. In comparison to Theorem 4, we obtain worse rates and it will be of interest to improve the rates of Theorem 5 while handling unbounded functions. ∎

## 5  Discussion

In this paper, we presented the first detailed theoretical analysis about the approximation quality of random Fourier features (RFF) that was proposed by [13] in the context of improving the computational complexity of kernel machines. While [13, 22] provided a probabilistic bound on the uniform approximation (over compact subsets of $\mathbb{R}^d$) of a kernel by random features, the result is not optimal. We improved this result by providing a finite-sample bound with optimal rate of convergence and also analyzed the quality of approximation in $L^r$-norm ($1 \leq r < \infty$). We also proposed an RFF approximation for derivatives of a kernel and provided theoretical guarantees on the quality of approximation in uniform and $L^r$-norms over compact subsets of $\mathbb{R}^d$.

While all the results in this paper (and also in the literature) dealt with the approximation quality of RFF over only compact subsets of $\mathbb{R}^d$, it is of interest to understand its behavior over entire $\mathbb{R}^d$. However, as discussed in Remark 1*(ii)* and in the paragraph following Theorem 3, RFF cannot approximate the kernel uniformly or in $L^r$-norm over $\mathbb{R}^d$. By truncating the Taylor series expansion of the exponential function, [3] proposed a non-random finite dimensional representation to approximate the Gaussian kernel which also enjoys the computational advantages of RFF. However, this representation also does not approximate the Gaussian kernel uniformly over $\mathbb{R}^d$. Therefore, the question remains whether it is possible to approximate a kernel uniformly or in $L^r$-norm over $\mathbb{R}^d$ but still retaining the computational advantages associated with RFF.

### Acknowledgments

Z. Szabó wishes to thank the Gatsby Charitable Foundation for its generous support.

## Footnotes

[1][22] derived tighter constants compared to [13] and also considered different RFF implementations.

[2]For example, in applications like kernel ridge regression based on RFF, it is more appropriate to consider the approximation guarantee in $L^2$ norm than in the uniform norm.

[3]We also correct some technical issues in the proof of [13, Claim 1], where (i) a shift-invariant argument was applied to the *non-shift invariant* kernel estimator $\hat{k}(\mathbf{x}, \mathbf{y}) = \frac{1}{m}\sum_{j=1}^m 2\cos(\boldsymbol{\omega}_j^T\mathbf{x} + b_j)\cos(\boldsymbol{\omega}_j^T\mathbf{y} + b_j) = \frac{1}{m}\sum_{j=1}^m \left[ \cos(\boldsymbol{\omega}_j^T(\mathbf{x} - \mathbf{y})) + \cos(\boldsymbol{\omega}_j^T(\mathbf{x} + \mathbf{y}) + 2b_j) \right]$, (ii) the *convexity* of $\mathbb{S}$ was not imposed leading to possibly undefined Lipschitz constant ($L$) and (iii) the randomness of $\boldsymbol{\Delta}^* = \arg\max_{\boldsymbol{\Delta} \in \mathbb{S}_\Delta} \|\nabla[k(\boldsymbol{\Delta}) - \hat{k}(\boldsymbol{\Delta})]\|_2$ was not taken into account, thus the upper bound on the expectation of the squared Lipschitz constant ($\mathbb{E}[L^2]$) does not hold.

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
