[Supplementary Material · nips2015_supplement.pdf]

# Supplement

## A Definitions & notation

Let $(Z, \rho)$ be a metric space, $(\Omega, \mathcal{A})$ a measurable space and $L_0(\Omega, \mathcal{A})$ denotes the set of $(\Omega, \mathcal{A}) \mapsto \mathbb{R}$ measurable functions. A family of maps $\mathcal{G} = \{g_z\}_{z \in Z} \subseteq L_0(\Omega, \mathcal{A})$ is called a separable Carathéodory family w.r.t. $Z$ if $(Z, \rho)$ is separable and $z \mapsto g_z(\omega)$ is continuous for all $\omega \in \Omega$. Let $\mathcal{G} \subseteq L_0(\Omega, \mathcal{A})$, $\boldsymbol{\varepsilon} = (\varepsilon_1, \ldots, \varepsilon_m)$ be a Rademacher sequence, i.e., $\varepsilon_j$-s are i.i.d. and $\mathbb{P}(\varepsilon_j = 1) = \mathbb{P}(\varepsilon_j = -1) = \frac{1}{2}$, and $(\omega_j)_{j=1}^m \in \Omega^m$. The Rademacher average of $\mathcal{G}$ is defined as $\mathcal{R}(\mathcal{G}, \boldsymbol{\omega}_{1:m}) :=$ $\mathbb{E}_{\boldsymbol{\varepsilon}} \sup_{g \in \mathcal{G}} \left| \frac{1}{m} \sum_{j=1}^m \varepsilon_j g(\omega_j) \right|$; we use the shorthand $\boldsymbol{\omega}_{1:m} = (\boldsymbol{\omega}_1, \ldots, \boldsymbol{\omega}_m)$. $S \subseteq Z$ is said to be an $r$-net of $Z$ if for any $z \in Z$ there is an $s \in S$ such that $\rho(s, z) \leq r$. The $r$-covering number of $Z$ is defined as the size of the smallest $r$-net, i.e., $\mathcal{N}(Z, \rho, r) = \inf \left\{ \ell \geq 1 : \exists s_1, \ldots, s_\ell \text{ such that } Z \subseteq \cup_{j=1}^\ell B_\rho(s_j, r) \right\}$, where $B_\rho(s, r) = \{z \in Z : \rho(z, s) \leq r\}$ is the closed ball with center $s \in Z$ and radius $r$. $\log \mathcal{N}(Z, \rho, r)$ is called the metric entropy. A $(Z, \|\cdot\|)$ Banach space is said to be of type $q \in (1, 2]$ if there exists a constant $C \in \mathbb{R}$ such that the $\mathbb{E}_{\boldsymbol{\varepsilon}} \left\| \sum_{j=1}^m \varepsilon_j f_j \right\| \leq C \left( \sum_{j=1}^m \|f_j\|^q \right)^{\frac{1}{q}}$ holds for every finite set of vectors $\{f_j\}_{j=1}^m \subseteq Z$. For example, $L^r(\Omega, \mathcal{A}, \mu)$ spaces are of type $q = \min(2, r)$ [6, page 73], where the $C$ constant only depends on $r$ ($C = C_r$). For a $(Z, \|\cdot\|)$ normed space, $Z^*$ denotes the space of continuous linear functionals on $Z$.

## B Proofs

We provide proofs of the results presented in Sections 3 and 4. Lemmas used in the proofs are enlisted in Section C.

### B.1 Proof of Theorems 1 and 4

Below we prove Theorem 4, thereby Theorem 1 ($\mathbf{p} = \mathbf{q} = \mathbf{0}$). The idea of the proof is as follows: *(i)* We note that

$$\|\partial^{\mathbf{p},\mathbf{q}} k - s^{\mathbf{p},\mathbf{q}}\|_{\mathcal{S} \times \mathcal{S}} = \sup_{\mathbf{x}, \mathbf{y} \in \mathcal{S}} |\partial^{\mathbf{p},\mathbf{q}} k(\mathbf{x}, \mathbf{y}) - s^{\mathbf{p},\mathbf{q}}(\mathbf{x}, \mathbf{y})| = \sup_{g \in \mathcal{G}} |\Lambda g - \Lambda_m g| =: \|\Lambda - \Lambda_m\|_{\mathcal{G}}, \tag{B.1}$$

where $\mathcal{G} := \{g_{\mathbf{z}} : \mathbf{z} \in \mathcal{S}_\Delta\}$ and $g_{\mathbf{z}} : \mathrm{supp}(\Lambda) \to \mathbb{R}$, $\boldsymbol{\omega} \mapsto \boldsymbol{\omega}^{\mathbf{p}}(-\boldsymbol{\omega})^{\mathbf{q}} h_{|\mathbf{p}+\mathbf{q}|}(\boldsymbol{\omega}^T \mathbf{z})$, which means the object of interest is the suprema of an empirical process indexed by $\mathcal{G}$. *(ii)* We show that $\|\Lambda - \Lambda_m\|_{\mathcal{G}}$ is measurable w.r.t. $\Lambda^m$ by verifying that $\mathcal{G}$ is a separable Carathéodory family (see the discussion following Definition 7.4 in [9]). *(iii)* (B.1) can be shown to satisfy the bounded difference property in C.1 and therefore by McDiarmid's inequality (Lemma C.1), $\|\Lambda - \Lambda_m\|_{\mathcal{G}}$ concentrates around its expectation. *(iv)* By applying the symmetrization lemma [9, Proposition 7.10] for the uniformly bounded function family $\mathcal{G}$, we obtain an upper bound in terms of the expected Rademacher average of $\mathcal{G}$. *(v)* The Rademacher average is bounded by the metric entropy of $\mathcal{G}$ (making use of the Dudley's entropy integral [2, Equation 4.4]), for which we can get an estimate by showing that $\mathcal{G}$ is a smoothly parametrized function class using the compactness of $\mathcal{S}_\Delta$.

- $\mathcal{G}$ **is a separable Carathéodory family:** $\mathcal{G}$ is a separable Carathéodory family w.r.t. $\mathcal{S}_\Delta$ since
  1. $g_{\mathbf{z}} : \mathrm{supp}(\Lambda) \to \mathbb{R}$, $\boldsymbol{\omega} \mapsto \boldsymbol{\omega}^{\mathbf{p}}(-\boldsymbol{\omega})^{\mathbf{q}} h_{|\mathbf{p}+\mathbf{q}|}(\boldsymbol{\omega}^T \mathbf{z})$ is measurable for all $\mathbf{z} \in \mathcal{S}_\Delta$.
  2. $\mathcal{S}_\Delta \subseteq \mathbb{R}^d$ is separable since $\mathbb{R}^d$ is separable.
  3. $\mathbf{z} \mapsto \boldsymbol{\omega}^{\mathbf{p}}(-\boldsymbol{\omega})^{\mathbf{q}} h_{|\mathbf{p}+\mathbf{q}|}(\boldsymbol{\omega}^T \mathbf{z})$ is continuous for all $\boldsymbol{\omega} \in \mathrm{supp}(\Lambda)$.

- **Concentration of** $\|\Lambda - \Lambda_m\|_{\mathcal{G}}$ **by its bounded difference property**: By defining $f(\boldsymbol{\omega}_1, \ldots, \boldsymbol{\omega}_m) := \|\Lambda - \Lambda_m\|_{\mathcal{G}}$, we have that for $\forall i \in \{1, \ldots, m\}$,

$$|f(\boldsymbol{\omega}_1, \ldots, \boldsymbol{\omega}_{i-1}, \boldsymbol{\omega}_i, \boldsymbol{\omega}_{i+1}, \ldots, \boldsymbol{\omega}_m) - f(\boldsymbol{\omega}_1, \ldots, \boldsymbol{\omega}_{i-1}, \boldsymbol{\omega}_i', \boldsymbol{\omega}_{i+1}, \ldots, \boldsymbol{\omega}_m)| =$$

$$= \left| \sup_{g \in \mathcal{G}} \left| \Lambda g - \frac{1}{m} \sum_{j=1}^m g(\boldsymbol{\omega}_j) \right| - \sup_{g \in \mathcal{G}} \left| \Lambda g - \frac{1}{m} \sum_{j=1}^m g(\boldsymbol{\omega}_j) + \frac{1}{m} [g(\boldsymbol{\omega}_i) - g(\boldsymbol{\omega}_i')] \right| \right| \leq \frac{1}{m} \sup_{g \in \mathcal{G}} |g(\boldsymbol{\omega}_i) - g(\boldsymbol{\omega}_i')|$$

$$\leq \frac{1}{m} \sup_{g \in \mathcal{G}} (|g(\boldsymbol{\omega}_i)| + |g(\boldsymbol{\omega}_i')|) \leq \frac{1}{m} \left[ \sup_{g \in \mathcal{G}} |g(\boldsymbol{\omega}_i)| + \sup_{g \in \mathcal{G}} |g(\boldsymbol{\omega}_i')| \right] \leq \frac{1}{m} [\boldsymbol{\omega}_i^{\mathbf{p}+\mathbf{q}} + |(\boldsymbol{\omega}_i')^{\mathbf{p}+\mathbf{q}}|] \leq \frac{2T_{\mathbf{p},\mathbf{q}}}{m}.$$

Applying McDiarmid's inequality (Lemma C.1) to $f$, for any $\tau > 0$, with probability at least $1 - e^{-\tau}$ over the choice of $(\boldsymbol{\omega}_i)_{i=1}^m \overset{i.i.d.}{\sim} \Lambda$,

$$\|\Lambda - \Lambda_m\|_{\mathcal{G}} \leq \mathbb{E}_{\boldsymbol{\omega}_{1:m}} \|\Lambda - \Lambda_m\|_{\mathcal{G}} + T_{\mathbf{p},\mathbf{q}} \sqrt{\frac{2\tau}{m}}. \tag{B.2}$$

- **Bounding $\mathbb{E}_{\boldsymbol{\omega}_{1:m}} \|\Lambda - \Lambda_m\|_{\mathcal{G}}$:** By the symmetrization lemma [9, Proposition 7.10] applied for the uniformly bounded function family $\mathcal{G}$ ($\sup_{g \in \mathcal{G}} \|g\|_\infty \le T_{\mathbf{p},\mathbf{q}} < \infty$), we have

$$\mathbb{E}_{\boldsymbol{\omega}_{1:m}} \|\Lambda - \Lambda_m\|_{\mathcal{G}} \le 2 \, \mathbb{E}_{\boldsymbol{\omega}_{1:m}} \mathcal{R}\left(\mathcal{G}, \boldsymbol{\omega}_{1:m}\right). \tag{B.3}$$

- **Bounding $\mathcal{R}\left(\mathcal{G}, \boldsymbol{\omega}_{1:m}\right)$:** Using Dudley's entropy integral [2, Equation 4.4], we have

$$\mathcal{R}\left(\mathcal{G}, \boldsymbol{\omega}_{1:m}\right) \le \frac{8\sqrt{2}}{\sqrt{m}} \int_0^{|\mathcal{G}|_{L^2(\Lambda_m)}} \sqrt{\log \mathcal{N}(\mathcal{G}, L^2(\Lambda_m), r)} \, \mathrm{d}r. \tag{B.4}$$

The upper limit of the integral can be bounded as

$$|\mathcal{G}|_{L^2(\Lambda_m)} = \sup_{g_1, g_2 \in \mathcal{G}} \|g_1 - g_2\|_{L^2(\Lambda_m)} \le \sup_{g_1, g_2 \in \mathcal{G}} \left(\|g_1\| + \|g_2\|_{L^2(\Lambda_m)}\right) \le 2 \sup_{g \in \mathcal{G}} \|g\|_{L^2(\Lambda_m)} \overset{(*)}{\le} 2\sqrt{T_{2\mathbf{p},2\mathbf{q}}}, \tag{B.5}$$

where $(*)$ follows from

$$\sup_{g \in \mathcal{G}} \|g\|_{L^2(\Lambda_m)} = \sup_{\mathbf{z} \in \mathcal{S}_\Delta} \sqrt{\frac{1}{m} \sum_{j=1}^m g_{\mathbf{z}}^2(\boldsymbol{\omega}_j)} = \sup_{\mathbf{z} \in \mathcal{S}_\Delta} \sqrt{\frac{1}{m} \sum_{j=1}^m \left[\boldsymbol{\omega}_j^{\mathbf{p}}(-\boldsymbol{\omega}_j)^{\mathbf{q}} h_{|\mathbf{p}+\mathbf{q}|}\left(\boldsymbol{\omega}_j^T \mathbf{z}\right)\right]^2} \le \sqrt{\frac{1}{m} \sum_{j=1}^m \boldsymbol{\omega}_j^{2(\mathbf{p}+\mathbf{q})}} \le \sqrt{T_{2\mathbf{p},2\mathbf{q}}}.$$

- **Bounding $\mathcal{N}(\mathcal{G}, L^2(\Lambda_m), r)$ by the compactness of $\mathcal{S}_\Delta$:** For any $g_{\mathbf{z}_1}, g_{\mathbf{z}_2} \in \mathcal{G}$,

$$\|g_{\mathbf{z}_1} - g_{\mathbf{z}_2}\|_{L^2(\Lambda_m)} = \left\|\boldsymbol{\omega} \mapsto \boldsymbol{\omega}^{\mathbf{p}}(-\boldsymbol{\omega})^{\mathbf{q}} \left(h_{|\mathbf{p}+\mathbf{q}|}\left(\boldsymbol{\omega}^T \mathbf{z}_1\right) - h_{|\mathbf{p}+\mathbf{q}|}\left(\boldsymbol{\omega}^T \mathbf{z}_2\right)\right)\right\|_{L^2(\Lambda_m)}.$$

By the mean value theorem, there exists $c \in (0,1)$ such that

$$\left|h_{|\mathbf{p}+\mathbf{q}|}\left(\boldsymbol{\omega}^T \mathbf{z}_1\right) - h_{|\mathbf{p}+\mathbf{q}|}\left(\boldsymbol{\omega}^T \mathbf{z}_2\right)\right| \le \left\|\nabla_{\mathbf{z}} h_{|\mathbf{p}+\mathbf{q}|}\left(\boldsymbol{\omega}^T (c\mathbf{z}_1 + (1-c)\mathbf{z}_2)\right)\right\|_2 \|\mathbf{z}_1 - \mathbf{z}_2\|_2 \,,$$

where

$$\left\|\nabla_{\mathbf{z}} h_{|\mathbf{p}+\mathbf{q}|}\left(\boldsymbol{\omega}^T (c\mathbf{z}_1 + (1-c)\mathbf{z}_2)\right)\right\|_2 \le \|\boldsymbol{\omega}\|_2.$$

Therefore,

$$\|g_{\mathbf{z}_1} - g_{\mathbf{z}_2}\|_{L^2(\Lambda_m)} \le \sqrt{\frac{1}{m} \sum_{j=1}^m \left(\left|\boldsymbol{\omega}_j^{\mathbf{p}+\mathbf{q}}\right| \|\boldsymbol{\omega}_j\|_2 \|\mathbf{z}_1 - \mathbf{z}_2\|_2\right)^2} = \|\mathbf{z}_1 - \mathbf{z}_2\|_2 \sqrt{\frac{1}{m} \sum_{j=1}^m \left|\boldsymbol{\omega}_j^{2(\mathbf{p}+\mathbf{q})}\right| \|\boldsymbol{\omega}_j\|_2^2}. \tag{B.6}$$

(B.6) shows that the existence of an $\epsilon$-net on $(\mathcal{S}_\Delta, \|\cdot\|_2)$ implies an $r = \epsilon \sqrt{\frac{1}{m} \sum_{j=1}^m \left|\boldsymbol{\omega}_j^{2(\mathbf{p}+\mathbf{q})}\right| \|\boldsymbol{\omega}_j\|_2^2}$-net on $(\mathcal{G}, L^2(\Lambda_m))$. In other words,

$$\mathcal{N}\left(\mathcal{G}, L^2(\Lambda_m), r\right) \le \mathcal{N}\left(\mathcal{S}_\Delta, \|\cdot\|_2, r \left(\frac{1}{m} \sum_{j=1}^m \left|\boldsymbol{\omega}_j^{2(\mathbf{p}+\mathbf{q})}\right| \|\boldsymbol{\omega}_j\|_2^2\right)^{-\frac{1}{2}}\right).$$

Define

$$A_{\mathbf{p},\mathbf{q}} := \sqrt{\frac{1}{m} \sum_{j=1}^m \left|\boldsymbol{\omega}_j^{2(\mathbf{p}+\mathbf{q})}\right| \|\boldsymbol{\omega}_j\|_2^2}.$$

By using the fact that $\mathcal{S}_\Delta \subseteq B_{\|\cdot\|_2}\left(\mathbf{t}, \frac{|\mathcal{S}_\Delta|}{2}\right)$ for some $\mathbf{t} \in \mathbb{R}^d$ and $\mathcal{N}(B_{\|\cdot\|_2}(\mathbf{s}, R), \|\cdot\|_2, \epsilon) \le \left(\frac{4R}{\epsilon} + 1\right)^d$ for any $\mathbf{s} \in \mathbb{R}^d$ [10, Lemma 2.5, page 20], we obtain

$$\mathcal{N}\left(\mathcal{G}, L^2(\Lambda_m), r\right) \le \left(\frac{4|\mathcal{S}|A_{\mathbf{p},\mathbf{q}}}{r} + 1\right)^d, \tag{B.7}$$

by noting that $|\mathcal{S}_\Delta| \le 2|\mathcal{S}|$. Using (B.5) and (B.7) in (B.4), we have

$$\mathcal{R}\left(\mathcal{G}, \boldsymbol{\omega}_{1:m}\right) \le \frac{8\sqrt{2d}}{\sqrt{m}} \int_0^{2\sqrt{T_{2\mathbf{p},2\mathbf{q}}}} \sqrt{\log\left(\frac{4|\mathcal{S}|A_{\mathbf{p},\mathbf{q}}}{r} + 1\right)} \, \mathrm{d}r \le \frac{8\sqrt{2d}}{\sqrt{m}} \int_0^{2\sqrt{T_{2\mathbf{p},2\mathbf{q}}}} \sqrt{\log\left(\frac{4|\mathcal{S}|A_{\mathbf{p},\mathbf{q}} + 2\sqrt{T_{2\mathbf{p},2\mathbf{q}}}}{r}\right)} \, \mathrm{d}r, \tag{B.8}$$

where in the last inequality we used the fact that $r \leq 2\sqrt{T_{2\mathbf{p},2\mathbf{q}}}$. By bounding $2|\mathcal{S}|A_{\mathbf{p},\mathbf{q}} + \sqrt{T_{2\mathbf{p},2\mathbf{q}}} \leq (2|\mathcal{S}| + \sqrt{T_{2\mathbf{p},2\mathbf{q}}})(A_{\mathbf{p},\mathbf{q}} + 1)$, (B.8) reduces to

$$
\begin{aligned}
\mathcal{R}\left(\mathcal{G}, \boldsymbol{\omega}_{1:m}\right) & \leq \frac{8\sqrt{2d}}{\sqrt{m}}\left(\int_0^{2\sqrt{T_{2\mathbf{p},2\mathbf{q}}}} \sqrt{\log \frac{2\left(2|\mathcal{S}| + \sqrt{T_{2\mathbf{p},2\mathbf{q}}}\right)}{r}}\,\mathrm{d}r + 2\sqrt{T_{2\mathbf{p},2\mathbf{q}} \log(A_{\mathbf{p},\mathbf{q}} + 1)}\right) \\
& = \frac{16\sqrt{2d}}{\sqrt{m}}\sqrt{T_{2\mathbf{p},2\mathbf{q}}}\left(\int_0^1 \sqrt{\log \frac{B_{\mathbf{p},\mathbf{q}} + 1}{r}}\,\mathrm{d}r + \sqrt{\log(A_{\mathbf{p},\mathbf{q}} + 1)}\right),
\end{aligned}
\tag{B.9}
$$

where the last equality is obtained by changing the variable of integration and defining $B_{\mathbf{p},\mathbf{q}} := \frac{2|\mathcal{S}|}{\sqrt{T_{2\mathbf{p},2\mathbf{q}}}}$. By applying Lemma C.2 to bound the integral in (B.9), we obtain

$$
\mathcal{R}\left(\mathcal{G}, \boldsymbol{\omega}_{1:m}\right) \leq \frac{16\sqrt{2d}}{\sqrt{m}}\sqrt{T_{2\mathbf{p},2\mathbf{q}}}\left(\sqrt{\log(B_{\mathbf{p},\mathbf{q}} + 1)} + \frac{1}{2\sqrt{\log(B_{\mathbf{p},\mathbf{q}} + 1)}} + \sqrt{\log(A_{\mathbf{p},\mathbf{q}} + 1)}\right).
\tag{B.10}
$$

- **Bounding the expectation of the Rademacher average**: From (B.10), we have

$$
\mathbb{E}_{\boldsymbol{\omega}_{1:m}} \mathcal{R}\left(\mathcal{G}, \boldsymbol{\omega}_{1:m}\right) \leq \frac{16\sqrt{2d}}{\sqrt{m}}\sqrt{T_{2\mathbf{p},2\mathbf{q}}}\left[\sqrt{\log(B_{\mathbf{p},\mathbf{q}} + 1)} + \frac{1}{2\sqrt{\log(B_{\mathbf{p},\mathbf{q}} + 1)}} + \sqrt{\log\left(\sqrt{C_{2\mathbf{p},2\mathbf{q}}} + 1\right)}\right],
\tag{B.11}
$$

which is obtained by repeated applications of Jensen's inequality to bound $\mathbb{E}_{\boldsymbol{\omega}_{1:m}} \sqrt{\log(A_{\mathbf{p},\mathbf{q}} + 1)} \leq \sqrt{\mathbb{E}_{\boldsymbol{\omega}_{1:m}} \log(A_{\mathbf{p},\mathbf{q}} + 1)} \leq \sqrt{\log(\mathbb{E}_{\boldsymbol{\omega}_{1:m}} A_{\mathbf{p},\mathbf{q}} + 1)}$ where $\mathbb{E}_{\boldsymbol{\omega}_{1:m}} A_{\mathbf{p},\mathbf{q}} \leq \sqrt{\frac{1}{m}\sum_{j=1}^m \mathbb{E}_{\boldsymbol{\omega}_j}\left[\left|\boldsymbol{\omega}_j^{2(\mathbf{p}+\mathbf{q})}\right| \|\boldsymbol{\omega}_j\|_2^2\right]} \leq \sqrt{C_{2\mathbf{p},2\mathbf{q}}}$.

- **Final bound**: Combining (B.2), (B.3) and (B.11) yields the result. $\qquad\square$

## B.2 Proof of Theorem 3

Below we prove Theorem 3: (i) We show that $f(\boldsymbol{\omega}_1, \ldots, \boldsymbol{\omega}_m) := \|k - \hat{k}\|_{L^r(\mathcal{S})}$ satisfies the bounded difference property, hence by the McDiarmid's inequality (Lemma C.1) it concentrates around its expectation $\mathbb{E}\|k - \hat{k}\|_{L^r(\mathcal{S})}$. (ii) By $L^r(\mathcal{S}) = \left[L^{\tilde{r}}(\mathcal{S})\right]^*$ ($\frac{1}{r} + \frac{1}{\tilde{r}} = 1$), the separability of $L^{\tilde{r}}(\mathcal{S})$ and the symmetrization lemma [11, Lemma 2.3.1] the value of $\mathbb{E}\|k - \hat{k}\|_{L^r(\mathcal{S})}$ is upper bounded in terms of $\mathbb{E}_{\boldsymbol{\varepsilon}} \|\sum_{i=1}^m \varepsilon_i \cos(\langle \boldsymbol{\omega}_i, \cdot - \cdot \rangle)\|_{L^r(\mathcal{S})}$. (iii) Exploiting that $L^r(\mathcal{S})$ is of type $\min(r, 2)$ with a constant independent of $\mathcal{S}$, we get the result.

- **Concentration of $\|k - \hat{k}\|_{L^r(\mathcal{S})}$ by its bounded difference property**: Define $\hat{k}_i(\mathbf{x}, \mathbf{y}) = \frac{1}{m}\sum_{j \neq i} \cos(\boldsymbol{\omega}_j^T(\mathbf{x} - \mathbf{y})) + \frac{1}{m}\cos(\tilde{\boldsymbol{\omega}}_i^T(\mathbf{x} - \mathbf{y}))$ where $\tilde{\boldsymbol{\omega}}_i$ is an i.i.d. copy of $\boldsymbol{\omega}_i$. Then $\|k - \hat{k}\|_{L^r(\mathcal{S})}$ satisfies the bounded difference property in (C.1):

$$
\sup_{(\boldsymbol{\omega}_i)_{i=1}^m, \tilde{\boldsymbol{\omega}}_i} \left|\|k - \hat{k}\|_{L^r(\mathcal{S})} - \|k - \hat{k}_i\|_{L^r(\mathcal{S})}\right| \leq \sup_{(\boldsymbol{\omega}_i)_{i=1}^m, \tilde{\boldsymbol{\omega}}_i} \|\hat{k}_i - \hat{k}\|_{L^r(\mathcal{S})} \leq \frac{2}{m}\sup_{\boldsymbol{\omega}_i} \|\cos(\langle \boldsymbol{\omega}_i, \cdot - \cdot \rangle)\|_{L^r(\mathcal{S})} \leq \frac{2}{m}\mathrm{vol}^{2/r}(\mathcal{S})
$$

and therefore by McDiarmid's inequality (Lemma C.1), for any $\tau > 0$, with probability at least $1 - e^{-\tau}$ over the choice of $(\boldsymbol{\omega}_i)_{i=1}^m \sim \Lambda$, we have

$$
\|k - \hat{k}\|_{L^r(\mathcal{S})} \leq \mathbb{E}_{\boldsymbol{\omega}_{1:m}}\|k - \hat{k}\|_{L^r(\mathcal{S})} + \mathrm{vol}^{2/r}(\mathcal{S})\sqrt{\frac{2\tau}{m}}.
\tag{B.12}
$$

- **Symmetrization, reduction to $\mathbb{E}_{\boldsymbol{\varepsilon}} \|\sum_{i=1}^m \varepsilon_i \cos(\langle \boldsymbol{\omega}_i, \cdot - \cdot \rangle)\|_{L^r(\mathcal{S})}$**: Let $\tilde{r}$ be the dual exponent of $r$, in other words $\frac{1}{r} + \frac{1}{\tilde{r}} = 1$. Then, by $L^r(\mathcal{S}) = \left[L^{\tilde{r}}(\mathcal{S})\right]^*$ and the separability of $L^{\tilde{r}}(\mathcal{S})$, there exists (see Lemma C.4) a countable $\mathcal{G} \subseteq L^{\tilde{r}}(\mathcal{S})$ ($\forall g \in \mathcal{G}$, $\|g\|_{L^{\tilde{r}}(\mathcal{S})} = 1$) such that

$$
\|k - \hat{k}\|_{L^r(\mathcal{S})} = \sup_{g \in \mathcal{G}}\left|\int_{\mathcal{S} \times \mathcal{S}} g(\mathbf{x}, \mathbf{y})\left[k(\mathbf{x}, \mathbf{y}) - \hat{k}(\mathbf{x}, \mathbf{y})\right]\mathrm{d}\mathbf{x}\mathrm{d}\mathbf{y}\right|.
\tag{B.13}
$$

One can rewrite the argument of this supremum by Eqs. (1)-(2) as

$$
\begin{aligned}
\int_{\mathcal{S} \times \mathcal{S}} g(\mathbf{x}, \mathbf{y})\left[k(\mathbf{x}, \mathbf{y}) - \hat{k}(\mathbf{x}, \mathbf{y})\right]\mathrm{d}\mathbf{x}\mathrm{d}\mathbf{y} & = \int_{\mathcal{S} \times \mathcal{S}} g(\mathbf{x}, \mathbf{y})\left[\int_{\mathbb{R}^d} \cos(\boldsymbol{\omega}^T(\mathbf{x} - \mathbf{y}))\mathrm{d}(\Lambda - \Lambda_m)(\boldsymbol{\omega})\right]\mathrm{d}\mathbf{x}\mathrm{d}\mathbf{y} \\
& = \int_{\mathbb{R}^d}\left[\int_{\mathcal{S} \times \mathcal{S}} g(\mathbf{x}, \mathbf{y})\cos(\boldsymbol{\omega}^T(\mathbf{x} - \mathbf{y}))\mathrm{d}\mathbf{x}\mathrm{d}\mathbf{y}\right]\mathrm{d}(\Lambda - \Lambda_m)(\boldsymbol{\omega}),
\end{aligned}
$$

and thus

$$\|k - \hat{k}\|_{L^r(\mathbb{S})} = \sup_{\tilde{g} \in \tilde{\mathcal{G}}} |(\Lambda - \Lambda_m)\tilde{g}|, \tag{B.14}$$

where $\tilde{\mathcal{G}} := \{\tilde{g}_g : g \in \mathcal{G}\}$, $\tilde{g}_g(\boldsymbol{\omega}) = \int_{\mathbb{S} \times \mathbb{S}} g(\mathbf{x}, \mathbf{y}) \cos(\boldsymbol{\omega}^T(\mathbf{x} - \mathbf{y})) d\mathbf{x} d\mathbf{y}$ and $\tilde{g}_g$ is continuous. Hence, using (B.14) with the symmetrization lemma [11, Lemma 2.3.1] and (B.13), we have

$$\mathbb{E}_{\boldsymbol{\omega}_{1:m}} \|k - \hat{k}\|_{L^r(\mathbb{S})} \leq 2\mathbb{E}_{\boldsymbol{\omega}_{1:m}} \mathbb{E}_{\boldsymbol{\varepsilon}} \sup_{\tilde{g} \in \tilde{\mathcal{G}}} \left| \frac{1}{m} \sum_{i=1}^m \varepsilon_i \tilde{g}(\boldsymbol{\omega}_i) \right| = \frac{2}{m} \mathbb{E}_{\boldsymbol{\omega}_{1:m}} \mathbb{E}_{\boldsymbol{\varepsilon}} \sup_{g \in \mathcal{G}} \left| \sum_{i=1}^m \varepsilon_i \int_{\mathbb{S} \times \mathbb{S}} g(\mathbf{x}, \mathbf{y}) \cos\left(\boldsymbol{\omega}_i^T(\mathbf{x} - \mathbf{y})\right) d\mathbf{x} d\mathbf{y} \right|$$

$$= \frac{2}{m} \mathbb{E}_{\boldsymbol{\omega}_{1:m}} \mathbb{E}_{\boldsymbol{\varepsilon}} \sup_{g \in \mathcal{G}} \left| \int_{\mathbb{S} \times \mathbb{S}} g(\mathbf{x}, \mathbf{y}) \left[ \sum_{i=1}^m \varepsilon_i \cos\left(\boldsymbol{\omega}_i^T(\mathbf{x} - \mathbf{y})\right) \right] d\mathbf{x} d\mathbf{y} \right| = \frac{2}{m} \mathbb{E}_{\boldsymbol{\omega}_{1:m}} \mathbb{E}_{\boldsymbol{\varepsilon}} \left\| \sum_{i=1}^m \varepsilon_i \cos(\langle \boldsymbol{\omega}_i, \cdot - \cdot \rangle) \right\|_{L^r(\mathbb{S})} \tag{B.15}$$

where $(\varepsilon_i)_{i=1}^m$ is a Rademacher sequence and $\mathbb{E}_{\boldsymbol{\varepsilon}}$ is the conditional expectation w.r.t. $(\varepsilon_i)_{i=1}^m$ with $(\boldsymbol{\omega}_i)_{i=1}^m$ being the conditioning random variables. Notice that the measurability of $\tilde{g}_g$-s with the countable cardinality of $\tilde{\mathcal{G}}$ enabled us to write expectations instead of outer expectations in [11, Lemma 2.3.1, page 108-110], and hence in Eq. (B.15).

- **Bounding** $\mathbb{E}_{\varepsilon} \left\| \sum_{i=1}^m \varepsilon_i \cos(\langle \boldsymbol{\omega}_i, \cdot - \cdot \rangle) \right\|_{L^r(\mathbb{S})}$ **by the type of** $L^r(\mathbb{S})$:

$$\mathbb{E}_{\boldsymbol{\varepsilon}} \left\| \sum_{i=1}^m \varepsilon_i \cos(\langle \boldsymbol{\omega}_i, \cdot - \cdot \rangle) \right\|_{L^r(\mathbb{S})} \overset{(*)}{\leq} C_r' \left( \sum_{i=1}^m \| \cos(\langle \boldsymbol{\omega}_i, \cdot - \cdot \rangle) \|_{L^r(\mathbb{S})}^{\min\{r,2\}} \right)^{\frac{1}{\min\{r,2\}}} \leq C_r' \mathrm{vol}^{2/r}(\mathbb{S}) m^{\max\{\frac{1}{2}, \frac{1}{r}\}}, \tag{B.16}$$

since $L^r(\mathbb{S})$ is of type $\min(2, r)$ [6, page 73] and there exists a universal constant $C_r'$ independent of $\mathbb{S}$ (the so-called Khintchine constant) [5, page 247] such that $(*)$ holds; in addition we used

$$\sum_{i=1}^m \| \cos(\langle \boldsymbol{\omega}_i, \cdot - \cdot \rangle) \|_{L^r(\mathbb{S})}^{\min\{2,r\}} = \sum_{i=1}^m \left( \int_{\mathbb{S} \times \mathbb{S}} \left| \cos(\boldsymbol{\omega}_i^T(\mathbf{x} - \mathbf{y})) \right|^r d\mathbf{x} d\mathbf{y} \right)^{\frac{\min\{2,r\}}{r}} \leq m \left[ \mathrm{vol}^2(\mathbb{S}) \right]^{\frac{\min\{2,r\}}{r}},$$

and $\frac{1}{\min\{2,r\}} = \max\left\{\frac{1}{2}, \frac{1}{r}\right\}$.

Combining (B.12)–(B.16) and using the bound on $\mathrm{vol}(\mathbb{S})$ given in the proof of Corollary 2 yields the result. $\qquad \square$

## B.3 Proof of Theorem 5

Below we give the detailed proof of Theorem 5. At high-level the proof goes as follows: *(i)* By the compactness of $\mathbb{S}_\Delta$ (implied by that of $\mathbb{S}$) one can take an $r$-net covering $\mathbb{S}_\Delta$ (for any $r > 0$). *(ii)* Small approximation error can be guaranteed at the centers of the $r$-net by Bernstein's inequality combined with a union bound. *(iii)* Propagation of the error from the centers to arbitrary points is achieved by Lipschitzness. *(iv)* The Lipschitz constant is, however, a random quantity and we show with high probability that it is 'not too large'. *(v)* Union bounding the two events (small errors at the centers and small Lipschitz constant) leads to a uniform bound for arbitrary $r$, which holds with high probability. *(vi)* Optimizing over $r$ gives the stated result.

Formally, the proof is as follows. Let us define

$$B_{\mathbf{p}, \mathbf{q}, \mathbb{S}} := \mathbb{E}_{\boldsymbol{\omega} \sim \Lambda} \left[ \sup_{\mathbf{z} \in conv(\mathbb{S}_\Delta)} \|\nabla_{\mathbf{z}} f(\mathbf{z}; \boldsymbol{\omega})\|_2 \right],$$

where $f(\mathbf{z}; \boldsymbol{\omega}) = \partial^{\mathbf{p}, \mathbf{q}} k(\mathbf{z}) - \boldsymbol{\omega}^{\mathbf{p}}(-\boldsymbol{\omega})^{\mathbf{q}} h_{|\mathbf{p}+\mathbf{q}|}(\boldsymbol{\omega}^T \mathbf{z})$. Let us notice that since $conv(\mathbb{S}_\Delta)$ is compact (by the compactness of $\mathbb{S}_\Delta$, implied by that of $\mathbb{S}$) and $\mathbf{z} \mapsto \|\nabla_{\mathbf{z}} f(\mathbf{z}; \boldsymbol{\omega})\|_2$ is continuous, the supremum inside the expectation in $B_{\mathbf{p}, \mathbf{q}, \mathbb{S}}$ is finite for any $\boldsymbol{\omega}$.

- **Covering of** $\mathbb{S}_\Delta$**:** By the compactness of $\mathbb{S}_\Delta$ there exist an $r$-net with at most

$$N = \left( \frac{2|\mathbb{S}_\Delta|}{r} + 1 \right)^d \leq \left( \frac{4|\mathbb{S}|}{r} + 1 \right)^d \tag{B.17}$$

balls covering $\mathbb{S}_\Delta$ [10, Lemma 2.5, page 20], where we used that $|\mathbb{S}_\Delta| \leq 2|\mathbb{S}|$. Let us denote the centers of this $r$-net by $\mathbf{c}_1, \ldots, \mathbf{c}_N$.

- **Bounding** $\bar{f}(\mathbf{b}; \boldsymbol{\omega}_{1:m}) - \bar{f}(\mathbf{a}; \boldsymbol{\omega}_{1:m})$, **where** $\mathbf{a}, \mathbf{b} \in \mathcal{S}_\Delta$; $\boldsymbol{\omega}_{1:m} = (\boldsymbol{\omega}_1, \ldots, \boldsymbol{\omega}_m)$ **is fixed:** Let

$$\bar{f}(\mathbf{z}; \boldsymbol{\omega}_{1:m}) = \frac{1}{m} \sum_{j=1}^m f(\mathbf{z}; \boldsymbol{\omega}_j) = \frac{1}{m} \sum_{j=1}^m \left[ \partial^{\mathbf{p},\mathbf{q}} k(\mathbf{z}) - \boldsymbol{\omega}_j^{\mathbf{p}} (-\boldsymbol{\omega}_j)^{\mathbf{q}} h_{|\mathbf{p}+\mathbf{q}|} \left( \boldsymbol{\omega}_j^T \mathbf{z} \right) \right].$$

$\mathbf{z} \mapsto \bar{f}(\mathbf{z}; \boldsymbol{\omega}_{1:m})$ is continuously differentiable since $\psi$ is so. Thus by the mean value theorem $\exists t \in (0,1)$ such that

$$\bar{f}(\mathbf{b}; \boldsymbol{\omega}_{1:m}) - \bar{f}(\mathbf{a}; \boldsymbol{\omega}_{1:m}) = \left\langle \nabla_{\mathbf{z}} \bar{f}(t\mathbf{a} + (1-t)\mathbf{b}; \boldsymbol{\omega}_{1:m}), \mathbf{b} - \mathbf{a} \right\rangle.$$

Hence by the Cauchy-Bunyakovsky-Schwarz inequality, we get

$$\left| \bar{f}(\mathbf{b}; \boldsymbol{\omega}_{1:m}) - \bar{f}(\mathbf{a}; \boldsymbol{\omega}_{1:m}) \right| \leq \left\| \nabla_{\mathbf{z}} \bar{f}(t\mathbf{a} + (1-t)\mathbf{b}; \boldsymbol{\omega}_{1:m}) \right\|_2 \left\| \mathbf{b} - \mathbf{a} \right\|_2 \leq \sup_{\mathbf{z} \in conv(\mathcal{S}_\Delta)} \left\| \nabla_{\mathbf{z}} \bar{f}(\mathbf{z}; \boldsymbol{\omega}_{1:m}) \right\|_2 \left\| \mathbf{b} - \mathbf{a} \right\|_2$$

$$=: L(\boldsymbol{\omega}_{1:m}) \left\| \mathbf{b} - \mathbf{a} \right\|_2, \tag{B.18}$$

where we used the compactness of $conv(\mathcal{S}_\Delta)$ (implied by that of $\mathcal{S}_\Delta$) and the continuity of the $\mathbf{z} \mapsto \left\| \nabla_{\mathbf{z}} \bar{f}(\mathbf{z}; \boldsymbol{\omega}_{1:m}) \right\|_2$ mapping to guarantee that $L(\boldsymbol{\omega}_{1:m})$ exists, and it is finite for any $\boldsymbol{\omega}_{1:m}$.

- **Bound on** $\mathbb{E}_{\boldsymbol{\omega}_1, \ldots, \boldsymbol{\omega}_m}[L(\boldsymbol{\omega}_{1:m})]$**:** Using the definition of $\bar{f}(\mathbf{z}; \boldsymbol{\omega}_{1:m})$, the linearity of differentiation, and the triangle inequality, we get

$$\left\| \nabla_{\mathbf{z}} \bar{f}(\mathbf{z}; \boldsymbol{\omega}_{1:m}) \right\|_2 = \left\| \nabla_{\mathbf{z}} \left[ \frac{1}{m} \sum_{j=1}^m f(\mathbf{z}; \boldsymbol{\omega}_j) \right] \right\|_2 = \left\| \frac{1}{m} \sum_{j=1}^m \nabla_{\mathbf{z}} f(\mathbf{z}; \boldsymbol{\omega}_j) \right\|_2 \leq \frac{1}{m} \sum_{j=1}^m \left\| \nabla_{\mathbf{z}} f(\mathbf{z}; \boldsymbol{\omega}_j) \right\|_2.$$

Therefore,

$$\sup_{\mathbf{z} \in conv(\mathcal{S}_\Delta)} \left\| \nabla_{\mathbf{z}} \bar{f}(\mathbf{z}; \boldsymbol{\omega}_{1:m}) \right\|_2 \leq \frac{1}{m} \sum_{j=1}^m \sup_{\mathbf{z} \in conv(\mathcal{S}_\Delta)} \left\| \nabla_{\mathbf{z}} f(\mathbf{z}; \boldsymbol{\omega}_j) \right\|_2$$

and

$$\mathbb{E}_{\boldsymbol{\omega}_{1:m}}[L(\boldsymbol{\omega}_{1:m})] = \mathbb{E}_{\boldsymbol{\omega}_{1:m}} \left[ \sup_{\mathbf{z} \in conv(\mathcal{S}_\Delta)} \left\| \nabla_{\mathbf{z}} f(\mathbf{z}; \boldsymbol{\omega}_{1:m}) \right\|_2 \right] \leq \frac{1}{m} \sum_{j=1}^m \mathbb{E}_{\boldsymbol{\omega}_{1:m}} \left[ \sup_{\mathbf{z} \in conv(\mathcal{S}_\Delta)} \left\| \nabla_{\mathbf{z}} f(\mathbf{z}; \boldsymbol{\omega}_j) \right\|_2 \right]$$

$$= \frac{1}{m} \sum_{j=1}^m B_{\mathbf{p},\mathbf{q},\mathcal{S}} = B_{\mathbf{p},\mathbf{q},\mathcal{S}}. \tag{B.19}$$

- **Bound on** $B_{\mathbf{p},\mathbf{q},\mathcal{S}}$**:** Note that

$$\sup_{\mathbf{z} \in conv(\mathcal{S}_\Delta)} \left\| \nabla_{\mathbf{z}} f(\mathbf{z}; \boldsymbol{\omega}) \right\|_2 = \sup_{\mathbf{z} \in conv(\mathcal{S}_\Delta)} \left\| \nabla_{\mathbf{z}} \left[ \partial^{\mathbf{p},\mathbf{q}} k(\mathbf{z}) - \boldsymbol{\omega}^{\mathbf{p}} (-\boldsymbol{\omega})^{\mathbf{q}} h_{|\mathbf{p}+\mathbf{q}|} \left( \boldsymbol{\omega}^T \mathbf{z} \right) \right] \right\|_2$$

$$\leq \sup_{\mathbf{z} \in conv(\mathcal{S}_\Delta)} \left( \left\| \nabla_{\mathbf{z}} \left[ \partial^{\mathbf{p},\mathbf{q}} k(\mathbf{z}) \right] \right\|_2 + \left\| \nabla_{\mathbf{z}} \left[ \boldsymbol{\omega}^{\mathbf{p}} (-\boldsymbol{\omega})^{\mathbf{q}} h_{|\mathbf{p}+\mathbf{q}|} \left( \boldsymbol{\omega}^T \mathbf{z} \right) \right] \right\|_2 \right)$$

$$\leq \sup_{\mathbf{z} \in conv(\mathcal{S}_\Delta)} \left\| \nabla_{\mathbf{z}} \left[ \partial^{\mathbf{p},\mathbf{q}} k(\mathbf{z}) \right] \right\|_2 + \sup_{\mathbf{z} \in conv(\mathcal{S}_\Delta)} \left\| \nabla_{\mathbf{z}} \left[ \boldsymbol{\omega}^{\mathbf{p}} (-\boldsymbol{\omega})^{\mathbf{q}} h_{|\mathbf{p}+\mathbf{q}|} \left( \boldsymbol{\omega}^T \mathbf{z} \right) \right] \right\|_2$$

$$= D_{\mathbf{p},\mathbf{q},\mathcal{S}} + \sup_{\mathbf{z} \in conv(\mathcal{S}_\Delta)} \left\| \nabla_{\mathbf{z}} \left[ \boldsymbol{\omega}^{\mathbf{p}} (-\boldsymbol{\omega})^{\mathbf{q}} h_{|\mathbf{p}+\mathbf{q}|} \left( \boldsymbol{\omega}^T \mathbf{z} \right) \right] \right\|_2. \tag{B.20}$$

By the homogenity of norms ($\|a\mathbf{v}\| = |a| \|\mathbf{v}\|$), the chain rule, and $|h_a(v)| \leq 1$ ($\forall a, \forall v$)

$$\left\| \nabla_{\mathbf{z}} \left[ \boldsymbol{\omega}^{\mathbf{p}} (-\boldsymbol{\omega})^{\mathbf{q}} h_{|\mathbf{p}+\mathbf{q}|} \left( \boldsymbol{\omega}^T \mathbf{z} \right) \right] \right\|_2 = |\boldsymbol{\omega}^{\mathbf{p}+\mathbf{q}}| \left\| h_{|\mathbf{p}+\mathbf{q}|+1} \left( \boldsymbol{\omega}^T \mathbf{z} \right) \boldsymbol{\omega} \right\|_2 \leq |\boldsymbol{\omega}^{\mathbf{p}+\mathbf{q}}| \|\boldsymbol{\omega}\|_2. \tag{B.21}$$

Combining Eq. (B.20) and (B.21) results in the bound

$$B_{\mathbf{p},\mathbf{q},\mathcal{S}} = \mathbb{E}_{\boldsymbol{\omega} \sim \Lambda} \left[ \sup_{\mathbf{z} \in conv(\mathcal{S}_\Delta)} \left\| \nabla_{\mathbf{z}} f(\mathbf{z}; \boldsymbol{\omega}) \right\|_2 \right] \leq D_{\mathbf{p},\mathbf{q},\mathcal{S}} + \mathbb{E}_{\boldsymbol{\omega} \sim \Lambda} \left[ |\boldsymbol{\omega}^{\mathbf{p}+\mathbf{q}}| \|\boldsymbol{\omega}\|_2 \right] = D_{\mathbf{p},\mathbf{q},\mathcal{S}} + E_{\mathbf{p},\mathbf{q}}. \tag{B.22}$$

- **Error propagation from the net centers:** We will use the following note to propagate the error from the net centers ($\mathbf{c}_j$, $j = 1, \ldots, N$) to an arbitrary $\mathbf{z} \in \mathcal{S}_\Delta$ point. Note: If $|\bar{f}(\mathbf{c}_j; \boldsymbol{\omega}_{1:m})| < \frac{\epsilon}{2}$ ($\forall j$) and $L(\boldsymbol{\omega}_{1:m}) < \frac{\epsilon}{2r}$, then

$$|\bar{f}(\mathbf{z}; \boldsymbol{\omega}_{1:m})| < \epsilon \quad (\forall \mathbf{z} \in \mathcal{S}_\Delta). \tag{B.23}$$

Indeed

$$\left| |\bar{f}(\mathbf{z}; \boldsymbol{\omega}_{1:m})| - \underbrace{|\bar{f}(\mathbf{c}_j; \boldsymbol{\omega}_{1:m})|}_{< \frac{\epsilon}{2}} \right| \leq |\bar{f}(\mathbf{z}; \boldsymbol{\omega}_{1:m}) - \bar{f}(\mathbf{c}_j; \boldsymbol{\omega}_{1:m})| \leq \underbrace{L(\boldsymbol{\omega}_{1:m})}_{< \frac{\epsilon}{2r}} \underbrace{\|\mathbf{z} - \mathbf{c}_j\|_2}_{\leq r} < \frac{\epsilon}{2},$$

where we used (B.18) and our assumptions in the note, thereby yielding (B.23).

- **Guaranteeing the conditions of** (B.23) **with high probability:**
  - Notice that $\mathbb{E}_{\boldsymbol{\omega}\sim\Lambda}[f(\mathbf{z};\boldsymbol{\omega})]=\mathbf{0}$ ($\forall\mathbf{z}$). Also since (7) holds, applying Bernstein's inequality for the individual $\mathbf{c}_j$ points (Lemma C.3; $\xi_n := f(\mathbf{c}_j;\boldsymbol{\omega}_n)$, $n=1,\ldots,m$; $S := \sqrt{m}\sigma$) gives that for any $\eta > 0$

$$\Lambda^m\left(|\bar{f}(\mathbf{c}_j;\boldsymbol{\omega}_{1:m})| \geq \frac{\eta\sigma}{\sqrt{m}}\right) \leq e^{-\frac{1}{2}\frac{\eta^2}{1+\frac{\eta L}{\sqrt{m}\sigma}}}. \tag{B.24}$$

Setting $\epsilon = \frac{2\eta\sigma}{\sqrt{m}}$, (B.24) is written as

$$\Lambda^m\left(|\bar{f}(\mathbf{c}_j;\boldsymbol{\omega}_{1:m})| < \frac{\epsilon}{2}\right) \geq 1 - e^{-\frac{1}{2}\frac{\left(\frac{\sqrt{m}\epsilon}{2\sigma}\right)^2}{1+\frac{\frac{\sqrt{m}\epsilon}{2\sigma}L}{\sqrt{m}\sigma}}} = 1 - e^{-\frac{m\epsilon^2}{8\sigma^2\left(1+\frac{\epsilon L}{2\sigma^2}\right)}}.$$

By union bounding ($j = 1, \ldots, N$), we get

$$\Lambda^m\left(\cap_{j=1}^N\left\{|\bar{f}(\mathbf{c}_j;\boldsymbol{\omega}_{1:m})| < \frac{\epsilon}{2}\right\}\right) \geq 1 - Ne^{-\frac{m\epsilon^2}{8\sigma^2\left(1+\frac{\epsilon L}{2\sigma^2}\right)}}. \tag{B.25}$$

  - **Condition** $L(\boldsymbol{\omega}_{1:m}) < \frac{\epsilon}{2r}$: Applying Markov's inequality to $L(\boldsymbol{\omega}_{1:m})$ (note that $L(\boldsymbol{\omega}_{1:m})$ is non-negative), for any $t > 0$, we obtain

$$\Lambda^m\left(L(\boldsymbol{\omega}_{1:m}) \geq t\right) \leq \frac{\mathbb{E}_{\boldsymbol{\omega}_1,\ldots,\boldsymbol{\omega}_m}[L(\boldsymbol{\omega}_{1:m})]}{t} \leq \frac{D_{\mathbf{p},\mathbf{q},\mathcal{S}} + E_{\mathbf{p},\mathbf{q}}}{t},$$

by invoking (B.19) and (B.22). Choosing $t = \frac{\epsilon}{2r}$, we have

$$\Lambda^m\left(L(\boldsymbol{\omega}_{1:m}) < \frac{\epsilon}{2r}\right) \geq 1 - \frac{2r}{\epsilon}(D_{\mathbf{p},\mathbf{q},\mathcal{S}} + E_{\mathbf{p},\mathbf{q}}). \tag{B.26}$$

- **Final bound for any** $r > 0$: By (B.25) and (B.26), and substituting the explicit form of $N$ in (B.17), we get

$$\Lambda^m\left(\sup_{\mathbf{z}\in\mathcal{S}_\Delta}|\bar{f}(\mathbf{z};\boldsymbol{\omega}_{1:m})| < \epsilon\right) \geq \Lambda^m\left(\left\{L(\boldsymbol{\omega}_{1:m}) < \frac{\epsilon}{2r}\right\}\bigcap\cap_{j=1}^N\left\{|\bar{f}(\mathbf{c}_j;\boldsymbol{\omega}_{1:m})| < \frac{\epsilon}{2}\right\}\right)$$

$$\geq 1 - \left(\frac{4|\mathcal{S}|}{r}+1\right)^d e^{-\frac{m\epsilon^2}{8\sigma^2\left(1+\frac{\epsilon L}{2\sigma^2}\right)}} - \frac{2r}{\epsilon}(D_{\mathbf{p},\mathbf{q},\mathcal{S}}+E_{\mathbf{p},\mathbf{q}}) \overset{(\dagger)}{\geq} 1 - c_* - \kappa_1 r^{-d} - \kappa_2 r, \tag{B.27}$$

where we invoked the

$$\left(\frac{4|\mathcal{S}|}{r}+1\right)^d = \left[2\left(\frac{\frac{4|\mathcal{S}|}{r}}{2}+\frac{1}{2}\right)\right]^d = 2^d\left(\frac{\frac{4|\mathcal{S}|}{r}}{2}+\frac{1}{2}\right)^d \overset{(\dagger)}{\leq} 2^d\frac{1}{2}\left[\left(\frac{4|\mathcal{S}|}{r}\right)^d+1^d\right] = 2^{d-1}\left[\left(\frac{4|\mathcal{S}|}{r}\right)^d+1\right]$$

Jensen's inequality in ($\dagger$), $c_* := 2^{d-1}e^{-\frac{m\epsilon^2}{8\sigma^2\left(1+\frac{\epsilon L}{2\sigma^2}\right)}}$, $\kappa_1 := 4^d|\mathcal{S}|^d c_*$ and $\kappa_2 := \frac{2}{\epsilon}(D_{\mathbf{p},\mathbf{q},\mathcal{S}}+E_{\mathbf{p},\mathbf{q}})$.

- **Matching the two terms to choose** $r$: Maximizing w.r.t. $r$ in (B.27)

$$f(r) = \kappa_1 r^{-d} + \kappa_2 r \Rightarrow f'(r) = \kappa_1(-d)r^{-d-1}+\kappa_2 = 0 \Rightarrow \frac{d\kappa_1}{\kappa_2} = r^{d+1}$$

we note that $r = \left(\frac{d\kappa_1}{\kappa_2}\right)^{\frac{1}{d+1}}$ maximizes it. Using this in (B.27), we have

$$\Lambda^m\left(\sup_{\mathbf{z}\in\mathcal{S}_\Delta}|\bar{f}(\mathbf{z};\boldsymbol{\omega}_{1:m})| \geq \epsilon\right) \leq c_* + \kappa_1\left(\frac{d\kappa_1}{\kappa_2}\right)^{-\frac{d}{d+1}}+\kappa_2\left(\frac{d\kappa_1}{\kappa_2}\right)^{\frac{1}{d+1}} = c_* + F_d\kappa_1^{\frac{1}{d+1}}\kappa_2^{\frac{d}{d+1}}$$

$$= 2^{d-1}e^{-\frac{m\epsilon^2}{8\sigma^2\left(1+\frac{\epsilon L}{2\sigma^2}\right)}} + F_d\left[2^{3d-1}|\mathcal{S}|^d e^{-\frac{m\epsilon^2}{8\sigma^2\left(1+\frac{\epsilon L}{2\sigma^2}\right)}}\right]^{\frac{1}{d+1}}\left[\frac{2}{\epsilon}(D_{\mathbf{p},\mathbf{q},\mathcal{S}}+E_{\mathbf{p},\mathbf{q}})\right]^{\frac{d}{d+1}}$$

$$= 2^{d-1}e^{-\frac{m\epsilon^2}{8\sigma^2\left(1+\frac{\epsilon L}{2\sigma^2}\right)}} + F_d 2^{\frac{4d-1}{d+1}}\left[\frac{|\mathcal{S}|(D_{\mathbf{p},\mathbf{q},\mathcal{S}}+E_{\mathbf{p},\mathbf{q}})}{\epsilon}\right]^{\frac{d}{d+1}}e^{-\frac{m\epsilon^2}{8(d+1)\sigma^2\left(1+\frac{\epsilon L}{2\sigma^2}\right)}},$$

where $F_d := d^{-\frac{d}{d+1}} + d^{\frac{1}{d+1}}$.

## B.4 Proof of bounded supp$(\Lambda) \Rightarrow$ (7)

We prove that the boundedness of supp$(\Lambda)$ implies that of $f$ [see (B.28)], specifically (7).

*Proof*: Indeed, let

$$f(\mathbf{z};\boldsymbol{\omega}) = \partial^{\mathbf{p},\mathbf{q}}k(\mathbf{z}) - \boldsymbol{\omega}^{\mathbf{p}}(-\boldsymbol{\omega})^{\mathbf{q}}h_{|\mathbf{p}+\mathbf{q}|}\left(\boldsymbol{\omega}^T\mathbf{z}\right) = \left[\int_{\mathbb{R}^d}\boldsymbol{\omega}^{\mathbf{p}}(-\boldsymbol{\omega})^{\mathbf{q}}h_{|\mathbf{p}+\mathbf{q}|}\left(\boldsymbol{\omega}^T\mathbf{z}\right)\mathrm{d}\Lambda(\boldsymbol{\omega})\right] - \boldsymbol{\omega}^{\mathbf{p}}(-\boldsymbol{\omega})^{\mathbf{q}}h_{|\mathbf{p}+\mathbf{q}|}\left(\boldsymbol{\omega}^T\mathbf{z}\right). \quad \text{(B.28)}$$

Applying the triangle inequality and $|h_a(v)| \le 1$ $(\forall a, \forall v)$ we have

$$|f(\mathbf{z};\boldsymbol{\omega})| \le \left|\int_{\mathbb{R}^d}\mathbf{s}^{\mathbf{p}}(-\mathbf{s})^{\mathbf{q}}h_{|\mathbf{p}+\mathbf{q}|}\left(\mathbf{s}^T\mathbf{z}\right)\mathrm{d}\Lambda(\mathbf{s})\right| + \left|\boldsymbol{\omega}^{\mathbf{p}}(-\boldsymbol{\omega})^{\mathbf{q}}h_{|\mathbf{p}+\mathbf{q}|}\left(\boldsymbol{\omega}^T\mathbf{z}\right)\right| \le \int_{\mathbb{R}^d}\left|\mathbf{s}^{\mathbf{p}}(-\mathbf{s})^{\mathbf{q}}h_{|\mathbf{p}+\mathbf{q}|}\left(\mathbf{s}^T\mathbf{z}\right)\right|\mathrm{d}\Lambda(\mathbf{s}) + \left|\boldsymbol{\omega}^{\mathbf{p}+\mathbf{q}}\right|$$

$$\le \int_{\mathbb{R}^d}\left|\mathbf{s}^{\mathbf{p}+\mathbf{q}}\right|\mathrm{d}\Lambda(\mathbf{s}) + \left|\boldsymbol{\omega}^{\mathbf{p}+\mathbf{q}}\right| = \int_{\mathrm{supp}(\Lambda)}\left|\mathbf{s}^{\mathbf{p}+\mathbf{q}}\right|\mathrm{d}\Lambda(\mathbf{s}) + \left|\boldsymbol{\omega}^{\mathbf{p}+\mathbf{q}}\right| \le 2\sup_{\mathbf{s}\in\mathrm{supp}(\Lambda)}\left|\mathbf{s}^{\mathbf{p}+\mathbf{q}}\right|.$$

$K := \sup_{\mathbf{s}\in\mathrm{supp}(\Lambda)}\left|\mathbf{s}^{\mathbf{p}+\mathbf{q}}\right|$ is finite since supp$(\Lambda)$ is bounded, thus $|f(\mathbf{z};\boldsymbol{\omega})|$ is bounded.

## C  Supplementary results

In this section, we present some technical results that are used in the proofs.

**Lemma C.1** (McDiarmid Inequality [7]). *Let $(X_i)_{i=1}^m$ be $\mathcal{X}$-valued independent random variables. Suppose $f : \mathcal{X}^m \to \mathbb{R}$ satisfies the bounded difference property,*

$$\sup_{u_1,\dots,u_m,u_r'\in\mathcal{X}}|f(u_1,\dots,u_m) - f(u_1,\dots,u_{r-1},u_r',u_{r+1},\dots,u_m)| \le c_r \quad (\forall r = 1,\dots,m). \quad \text{(C.1)}$$

*Then for any $\epsilon > 0$,*

$$\mathbb{P}\left(f(X_1,\dots,X_m) - \mathbb{E}\left[f(X_1,\dots,X_m)\right] \ge \epsilon\right) \le e^{-\frac{2\epsilon^2}{\sum_{r=1}^m c_r^2}}.$$

*Note: specifically, if $c = c_r$ $(\forall r)$ then applying a $\tau = \frac{2\epsilon^2}{\sum_{r=1}^m c_r^2} = \frac{2\epsilon^2}{mc^2} \Leftrightarrow \epsilon = c\sqrt{\frac{\tau m}{2}}$ reparameterization one gets $\mathbb{P}\left(f(X_1,\dots,X_m) < \mathbb{E}\left[f(X_1,\dots,X_m)\right] + c\sqrt{\frac{\tau m}{2}}\right) \ge 1 - e^{-\tau}.$*

**Lemma C.2.** *For $a > 1$, $\int_0^1 \sqrt{\log\frac{a}{\epsilon}}\,\mathrm{d}\epsilon \le \sqrt{\log a} + \frac{1}{2\sqrt{\log a}}$.*

*Proof.* By change of variables, we have $\int_0^1 \sqrt{\log\frac{a}{\epsilon}}\,\mathrm{d}\epsilon = a\int_{\log a}^\infty \sqrt{t}e^{-t}\,\mathrm{d}t$. Applying partial integration, we have

$$\int_{\log a}^\infty \sqrt{t}e^{-t}\,\mathrm{d}t = [\sqrt{t}e^{-t}]_\infty^{\log a} + \int_{\log a}^\infty \frac{1}{2\sqrt{t}}e^{-t}\,\mathrm{d}t \le \frac{\sqrt{\log a}}{a} + \frac{1}{2\sqrt{\log a}}\int_{\log a}^\infty e^{-t}\,\mathrm{d}t = \frac{\sqrt{\log a}}{a} + \frac{1}{2a\sqrt{\log a}},$$

thereby yields the result. $\square$

**Lemma C.3** (Bernstein inequality [12]). *Let $\xi \in \mathbb{R}$ be a random variable, $\mathbb{E}_{\xi\sim\mathbb{P}}[\xi] = 0$, and assume that $\exists L > 0, S > 0$ satisfying*

$$\sum_{j=1}^m \mathbb{E}_{\xi_j\sim\mathbb{P}}\left[|\xi_j|^M\right] \le \frac{M!S^2L^{M-2}}{2} \quad (\forall M \ge 2),$$

*where $(\xi_j)_{j=1}^m \overset{i.i.d.}{\sim} \mathbb{P}$. Then for any $0 < m \in \mathbb{N}$, $\eta > 0$,*

$$\mathbb{P}^m\left(\left|\sum_{j=1}^m \xi_j\right| \ge \eta S\right) \le e^{-\frac{1}{2}\frac{\eta^2}{1+\frac{\eta L}{S}}}.$$

**Lemma C.4** ($L^r$ norm as countable supremum). *Assume that $1 < \tilde{r} < \infty$. If $(X, \mathcal{A}, \mu)$, $\mu(X) < \infty$, $\frac{1}{r} + \frac{1}{\tilde{r}} = 1$, then $\left[L^{\tilde{r}}(X, \mathcal{A}, \mu)\right]^* = \{F_f : f \in L^r(X, \mathcal{A}, \mu)\}$, where $F_f(u) = \int_X uf\mathrm{d}\mu$, and $\|f\|_{L^r} = \|F_f\|$ $(= \sup_{\|g\|_{L^{\tilde{r}}}=1}|F_f(g)|)$; see [8, Theorem 4.1]. Specifically, if $X = \mathbb{S} \subseteq \mathbb{R}^d$ compact and it is endowed with the Borel $\sigma$-algebra, then by the separability of $\mathbb{S}$, $L^{\tilde{r}}(\mathbb{S})$ is also separable [4, Prop. 3.4.5] since the Borel $\sigma$-algebra is countably generated [1, page 17 (vol. 2)], thus there exists a countable $\mathcal{G} \subseteq L^{\tilde{r}}(\mathbb{S})$, [3, Lemma 6.7] such that $\|g\|_{L^{\tilde{r}}(\mathbb{S})} = 1$ $(\forall g \in \mathcal{G})$ and $\|F_f\| = \sup_{g\in\mathcal{G}}|F_f(g)|$.*

*Note: the $\sigma$-algebra of Lebesgue measurable sets is typically* not *countably generated [1, page 106 (vol. I)].*