[Reviews · NeurIPS 2015]

Submitted by Assigned_Reviewer_1

The paper presents novel finite sample convergence bounds for random Fourier feature kernel approximations.

In particular shows an optimal O(log(|S|) / \sqrt{m}) uniform convergence bound, this improves upon the previously known bound O(|S| \sqrt{\log(m)/m}).

In particular, the dependence on the data diameter is significantly improved, and allows for the diameter to grow aggressively with m, while still maintaining convergence.

The authors also present bounds in terms of L_p norms, and analogous bounds on approximate kernel derivatives (where special care is required due to non-boundedness). Overall the paper is well written and the presented bounds are well discussed and compared.
Summary: The paper presents novel finite sample convergence bounds for random Fourier feature kernel approximations.

In particular shows an optimal O(log(|S|) / \sqrt{m}) uniform convergence bound, this improves upon the previously known bound O(|S| \sqrt{\log(m)/m}).

The paper also presents bounds in terms of general L_p norms and bounds on the approximation of the kernel derivatives. The results are well presented, discussed and compared.

Overall, I view the paper as a significant contribution.

Submitted by Assigned_Reviewer_2

Quality: the paper is rather well written.

Clarity: The introduction, notation and theorems are quite clear.

Originality: The contribution is a refinement of a bound in [12] plus new bounds for different norms. Moreover it introduces some bounds on the derivatives of the kernel.

Significance: While in the introduction was pointed out the importance of the random features, it is not clear the relevance of the improved bound, indeed the dependence on m is the same modulo logarithmic factor, while the dependence from another crucial factor, the input space dimension, seems the same in both bounds.

It is not clear to me what is the significance of the L^r bounds or their potential fields of application (the paper gives no hints about it). The result of the derivative of the kernel function seem new and to have potential applications.
Summary: The paper refines a bound for the uniform distance on compact subsets of the Random Feature approximation of the kernel function from the true kernel function. The new bound is of the order O(m^-1/2) instead of O(m^-1/2 log m) (m is the number of random features). Moreover the paper introduces other bounds for L^r distances between the two functions. Finally bounds on the derivatives of the kernel are introduced

While the result is clear and seems sound, It is not clear the relevance of the improved bound, because the dependence on m is just improved by a logarithmic factor and the dependence on the dimension of the input space, that is considered the bottleneck of random features w.r.t. e.g. nystrom methods, is the same. Moreover it is not clear the utility of the L^r bounds and the paper gives no hints about it.

The result on the derivative of the kernel function seem new and seem to have potential applications, as pointed out in the paper.

Submitted by Assigned_Reviewer_3

(this is a "light" review, I just fille the first box)
Summary: First, this paper is longer than all other papers I had to review. The margins have been reduced. This alone should be a motif for rejection.

This being said, the paper provides new estimates of the deviation between a translation-invariant kernel and its approximation by random features, uniform over compact sets. The estimates improve over the bounds provided in the original paper of Rahimi and Recht (NIPS 2007). However the authors do not cite the more recent paper of the same authors "Uniform approximation of functions with random bases" (http://dx.doi.org/10.1109/ALLERTON.2008.4797607) which seems also to provide tight bounds, using the proof technique of MacDiarmid etc.. It would be important to cite this paper and clarify how the submitted paper improves over it.

In terms of significance, this is a nice result to have the good rate (removing a logarithmic dependency), however it is unlikely to have a strong impact on the field.

### Added after author's rebuttal - OK for the format issue, I did not realize that the shorter margins came with less lines - I gave a second read to the paper (although this is a "light" review) and are more convinced now that there is truly some original and non-trivial contributions

Submitted by Assigned_Reviewer_4

The paper proposes optimal convergence rates for random fourier features (RFFs) in terms of RFF counts in uniform and Lr norm cases. In addition, they analyse the finite sample case and analyse the relation to growing the input diameter. The paper is an important contribution to RFFs. The paper is well written.

There's also analysis of kernel derivatives. Introducing RFFs for derivative features is a separate topic, which warrants also experimental results, and might have been better placed in a separate paper.

Minor comments: - notation A_delta = A - A = { x - y} is a bit strange - what is \Lambda^m?
Summary: The paper proposes optimal rates for fourier random features. The results are important in the field, properly derived and the paper is well written.

Submitted by Assigned_Reviewer_5

Given the attention that has been drawn to the random Fourier features, it is glad to see a paper trying to given a detailed analysis on how much features it needs to have descent performance. It is definitely an important problem as well to obtain better understanding of the kernel methods and the efforts to make it work for very large scale data set.

The paper is to provide optimal rates for the convergence of the kernel function in terms of the uniform and L^r norm. In the main results, the authors gave an improved a rate over the previous work: reducing the linear dependence of the diameter |S| to a logarithmic dependence.

This is definitely an important result for certain settings. However, I am more curious about the consistency of random Fourier features in the probabilistic setting given a probability measure, because in many real world problems, data are usually not sampled from a uniform distribution.

Moreover, I would also like to see some discussions about how the convergence of the kernel function will influence the performance of classification or other applications that use kernel method.

To sum up, the paper does deliver what the authors promise, even though I didn't carefully go through the proof of the main results. The results are interesting in its own right, but I feel its contribution to the literature is relatively incremental.
Summary: In this paper, the authors try to provide a detailed analysis to the widely used random Fourier features and claim to have the optimal converging rates in uniform and L^r norms.

Author Feedback
Author rebuttal: We thank the reviewers for their careful reading of the paper and valuable comments. In the following, we first address the common concerns and then the specific concerns raised by the reviewers.

1) Removing log(m) factor in the rate: Given the popularity of random Fourier features (RFF) in machine learning (ML), the primary focus of our paper is to lay down theoretical foundations for RFF in approximating the kernel (and its derivative) and derive approximation guarantees in terms of the number of RFFs (m) and the set size (|S|-diameter of S). Surprisingly, this fundamental question has only been partially addressed in the ML literature since the introduction of RFFs in 2007. In our work, we provided a complete answer to this fundamental question by improving uniform convergence rate from O(|S|\sqrt{log(m)/m}) to O(\sqrt{log|S|}/\sqrt{m}). While it appears that we have only shaved the extra log(m) term to obtain an optimal rate, the dependence on |S| is improved from linear to logarithmic. As discussed in detail in Remark 1(ii, iii), the logarithmic dependence on |S| is optimal and it ensures that the uniform convergence guarantee can be obtained not just over fixed compact set S but over entire R^d by growing |S| to infinity at an exponential rate, i.e., |S_m|=o(e^m) rather than at a sublinear rate, i.e., |S_m|=o(\sqrt{m/log(m)}). In other words, for the same approximation error, the kernel can be approximated uniformly over a significantly larger S than it was considered in the literature.

2) Convergence in L^r-norm: In addition to the convergence in uniform norm, we also provided approximation guarantees in L^r-norm. This is because the uniform norm is too strong and may not be suitable to attain correct rates in the generalization bounds associated with learning algorithms involving RFF. For example, in applications like kernel ridge regression based on RFF, it is more appropriate to consider the approximation guarantee in L^2 norm than in the uniform norm. In the revised version, we will clarify and elaborate on this motivation to study the approximation guarantee in L^r-norm.

3) Applications: We have not provided any theoretical analysis on the performance of RFF based learning algorithms. This is an interesting and challenging question and there exist a few papers in the literature that provide partial and suboptimal results. We are currently investigating this problem and have obtained few preliminary results. In the revised version, we will include a discussion about these existing results and possibly discuss how our results can be applied in RFF based learning algorithms.

Reviewer 2: Please refer to 1), 2) and 3) above.

Reviewer 3: Please refer to 3) above.

Reviewer 4: The notation A-A is quite standard and is the Minkowski difference between sets. \Lambda^m denotes the m-fold product measure of \Lambda (see the end of the 1st paragraph in Section 2).

Reviewer 5: We thank the reviewer for pointing us to "Uniform approximation of functions with random bases" (we refer to it as RR08). While RR08 is related to our paper, their focus is different. They considered approximation guarantees for random feature schemes in approximating a function that is a mixture of certain family of functions. But the RFF approach can be seen as a special case of the problem in RR08. While our rate of 1/\sqrt{m} seems to match with the one in Theorem 3.2 of RR08, there are some critical differences that highlight the importance of our result. First, the proof technique in RR08 only yields a linear dependence on |S| in comparison to the logarithmic dependence on |S| in our work. The advantage of log dependence is highlighted in 1) above. Second, Theorem 3.2 (in the present form) cannot handle RFF and requires some modification since the assumptions of Theorem 3.2 do not hold for the cosine function. Third, the analysis in RR08 assumes that the kernel is absolutely integrable, which is not required in our work. In addition, our work provides convergence in L^r-norms for the kernel and its derivative (see 2)), which is not covered in RR08.

We also thank the reviewer for pointing out the issue with reduced margin. In the tech report version of the paper we used \usepackage{geometry} in the preamble of the .tex file to (have commands to) control the margins, which we forgot to comment out in the NIPS version. This line seems to have the (unnoticed) side-effect of slightly reducing the margin on the left and right, and increasing it on the top and bottom. After commenting out this line, the recompiled submission (main + supplement) reduces from 9 pages to exactly 8 pages, each. This means our submitted version is not longer but in fact shorter though it appeared longer. However, we are very sorry for this error and we again thank the reviewer for raising our attention to this issue. Fixing it will enable us to include more detailed discussions [e.g., on 3)] in the revised version.